# Direction-specific enhanced diffusion of CO$_2$ in chiral hexagonal boron nitride nanotubes

Manh-Thuong Nguyen [1] ✉, David J. Heldebrant [1,2] ✉, Jian Liu [1], Abhoyjit S. Bhown[3] & Zhijie Xu [1]

To meet performance requirements, the next generation of gas separation membranes will need *both* high gas permeability and selectivity, attainable if we could coax adsorbates to minimize random Brownian motion and produce direction-specific diffusion along a desired axis. In this atomistic modeling study, we detail how direction-specific diffusion of CO$_2$ can be achieved in chiral hexagonal boron nitride nanotubes (hBNNTs) by means of a non-Knudsen diffusion mechanism. Our findings detail how this mechanism of diffusion is driven by interactions with the tube walls and enables the CO$_2$ molecules to diffuse along the nanotube's z-axis with minimized collisions and directional changes. hBNNTs with chiral indices exhibit CO$_2$ diffusion rates faster than non-chiral tubes of comparable and larger diameters. Of the hBNNTs studied, a (7,3) tube appears to be ideally sized (3.7 Å radius) exhibiting CO$_2$ diffusion that is 3.4 times faster than diatomic N$_2$. Applying this mechanism of diffusion to hypothetical sheet membranes prepared with aligned chiral (7,3) hBNNTs results in membranes with a calculated CO$_2$/N$_2$ permselectivity of 170 and a CO$_2$ permeability limit of nearly 1.35 ×10$^7$ Barrer, readily surpassing the Robeson upper bound for CO$_2$/N$_2$ separations.

Rotation is a common, omnipresent phenomenon that spans from massive neutron stars to spin in a magnetic field. Rotation can either enhance[1] or hinder[2] the transport of an object, depending on how the object interacts with its surrounding medium and other factors. Precession is a special, compound form of rotation that occurs when a spinning object experiences a torque that causes its axis to change direction in a slow, circular motion. Engineering recognizes various observable manifestations of precession, such as the stabilized trajectory of a spinning projectile and the resonance frequency shift in magnetic resonance imaging. Intriguingly, researchers have observed strong mass transport enhancement from miniscule precession[3], indicating its applicability as a potential strategy for controlling transport processes. Although introducing precession to objects has led to many advancements, its application to molecules in chemical processes has been limited. This could be because linearly symmetrical molecules like CO$_2$ cannot "rotate" about their own axis—molecular precession would only be possible if the molecules become slightly bent.

The motion of molecules is key to the gas-phase separations identified as potentially world-changing and important to industries[4–7]. Separations are one of the most common and energy-intensive chemical processes, often using specially designed adsorbents or membranes. Developing next-generation materials and processes for reducing the energy demand and costs for gas-phase separations requires innovative approaches to overcome conventional molecular phenomena. In traditional adsorbents, gases like CO$_2$ move via Brownian motion, a chaotic process where molecules flip, tumble, and bounce off interfaces or other molecules at random. Each potential change in direction makes the separation slower and less efficient than if all the molecules moved directionally through a sorbent or membrane.

We posited that we could achieve more efficient separations if we could prevent tumbling and collisions by coaxing molecules into a precession-like motion. Such movement could keep CO$_2$ oriented linearly along the pore axis instead of other directions that would

[1]Pacific Northwest National Laboratory, Richland, WA, USA. [2]Washington State University, Pullman, WA, USA. [3]Electric Power Research Institute, Palo Alto, CA, USA. ✉e-mail: manhthuong.nguyen@pnnl.gov; david.heldebrant@pnnl.gov

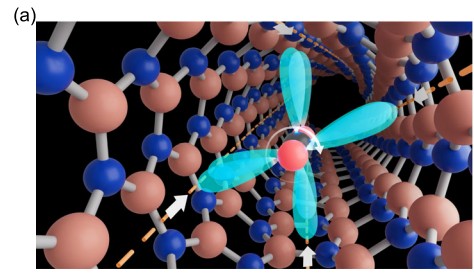

(a)

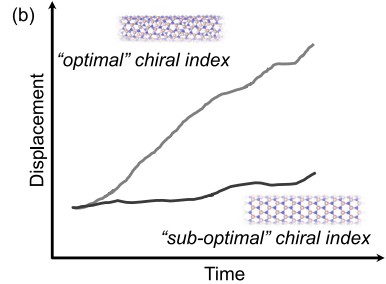

(b)

**Fig. 1 | The inspiration for this study. a** $CO_2$ precessing down the z-axis of a chiral hBNNT to avoid collisions. **b** Scheme shows what was discovered: enhanced transport can be achieved via tunable interactions between the substrate and the geometric orientation of hexagonal patterns in the wall of nanotubes. Colour code: B in pink and N in blue.

increase interactions with the wall and impede axial diffusion. Physical sorbents or membranes with well-defined straight pore channels are preferred targets for imparting molecular precession due to their finely tunable molecular geometries and lack of chemical fixation that would allow $CO_2$ to freely rotate inside.

To design a framework that could introduce molecular precession, we took inspiration from systems where the geometric orientation of a host enhances diffusion and directionality in a transient item. We posited that molecules of $CO_2$ could be coaxed into precessing down the horizontal axis of an appropriately sized chiral single-walled nanotube (Fig. 1a). If successful, this would minimize collisions by maintaining the molecule's orientation along the pore axis and enable rapid, directional diffusion along the Z-axis of the nanotube.

Promoting molecular precession requires two primary design criteria: a dipole and an external field. As $CO_2$ is a linear quadrupolar molecule, its $D_{\infty h}$ symmetry must first be broken to create a dipole that can "precess" in response to an external electric field. There have been reports of $CO_2$ molecules being deformed/distorted within nanometer-sized cavities[8,9], resulting in two π bonding orbitals rotated 90° from each other that can act as "fins" to be manipulated via electron repulsion to introduce precession-like motion (Fig. 1a). To satisfy the second criterion, we need to create a continuous rotating electric field for the dipole of the distorted $CO_2$ molecule to interact with. hBNNTs can be made with chiral indices and electron-rich nitrogen atoms (Fig. 1a, blue) that create a continuous weak electric field that follows the chiral index of the tube. This field could interact with $CO_2$ via electron repulsion to potentially induce precession-like motion. We thus chose to computationally study hBNNT tubes as they meet both of our design requirements: they can break the symmetry of $CO_2$ and present a continuous rotating electric field that could potentially coax the distorted molecule into motion that minimizes collisions and directional changes (Fig. 1a). What we found is the first ever evidence that diffusion rates are directly influenced by orientation of hexagonal patterns of boron nitride nanotubes (Fig. 1b).

Here, we detail a machine learning interatomic potential molecular dynamics (MLIPMD) study that describes a type of active motion of $CO_2$, representing, to our knowledge, the first-ever direction-specific diffusion of $CO_2$ down a desired axis of a sorbent. We will show that this motion can be achieved via coupled interactions of $CO_2$ with specific orientations of the hexagonal patterns of the walls of boron nitride nanotubes. Our findings suggest that we can tune these molecular-level interactions via the tube's chiral indices to minimize the collisions that determine the mean free path of diffusion to govern macroscopic transport. This fundamental phenomenon can occur for $CO_2$ (and likely other molecules) under nanoconfinement inside chiral nanotubes, enabling amplified rates of diffusion over non-chiral tubes of similar dimensions while retaining high selectivity. We close with a discussion of how these principles could be introduced to membranes and porous materials to design advanced materials that could exceed

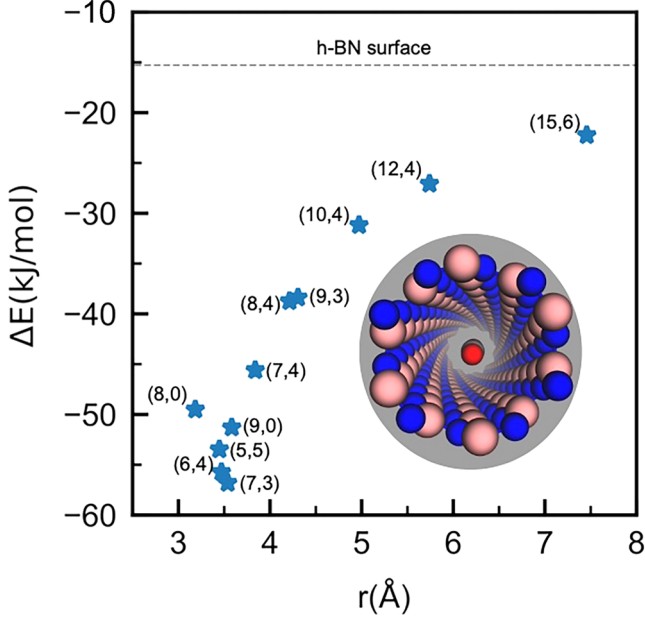

**Fig. 2 | Adsorption energy of $CO_2$ in hBNNTs.** Colour code: B in pink, N in blue, O in red, and C in grey.

or shift the Robeson upper bound for membrane-based $CO_2/N_2$ separations.

## Results and discussion

### Sizing the optimal tube radius for $CO_2$ adsorption

We selected single-walled hBNNTs with commonly made geometries, including armchair, zigzag, and chiral tubes[10]. We first screened the hBNNTs to find optimal dimensions by identifying tubes with a favorable calculated $CO_2$ binding energy, Fig. 2. The tube radius, from 3.18 Å ((8,0)) to 7.46 Å ((15,6)), Fig. S1, offers a range of $CO_2$–tube interaction scenarios. The binding energy of $CO_2$ on a flat h-BN sheet is calculated to be −15.3 kJ/mol, see "Methods" and Fig. S1. This weak interaction, consistent with the literature[11], indicates that the van der Waals interactions play a key role in the adsorption strength of $CO_2$ on the h-BN surface. The binding energy increases for nanotubes relative to a flat surface due to curvature, which raises the density of B and N atoms surrounding the $CO_2$. This becomes more obvious upon further decreasing the tube radius. Of the tubes under study (indices provided in Fig. 2), the (7,3) tube interacts most strongly with $CO_2$, with a binding energy of −57 kJ/mol ("−" implies attraction). The radius of this tube, 3.54 Å, is near the equilibrium distance of $CO_2$ on the h-BN surface, 3.55 Å. $CO_2$ is very close to the wall in tubes with smaller radii, resulting in Pauli repulsions and weakening of the $CO_2$-hBNNT bonding.

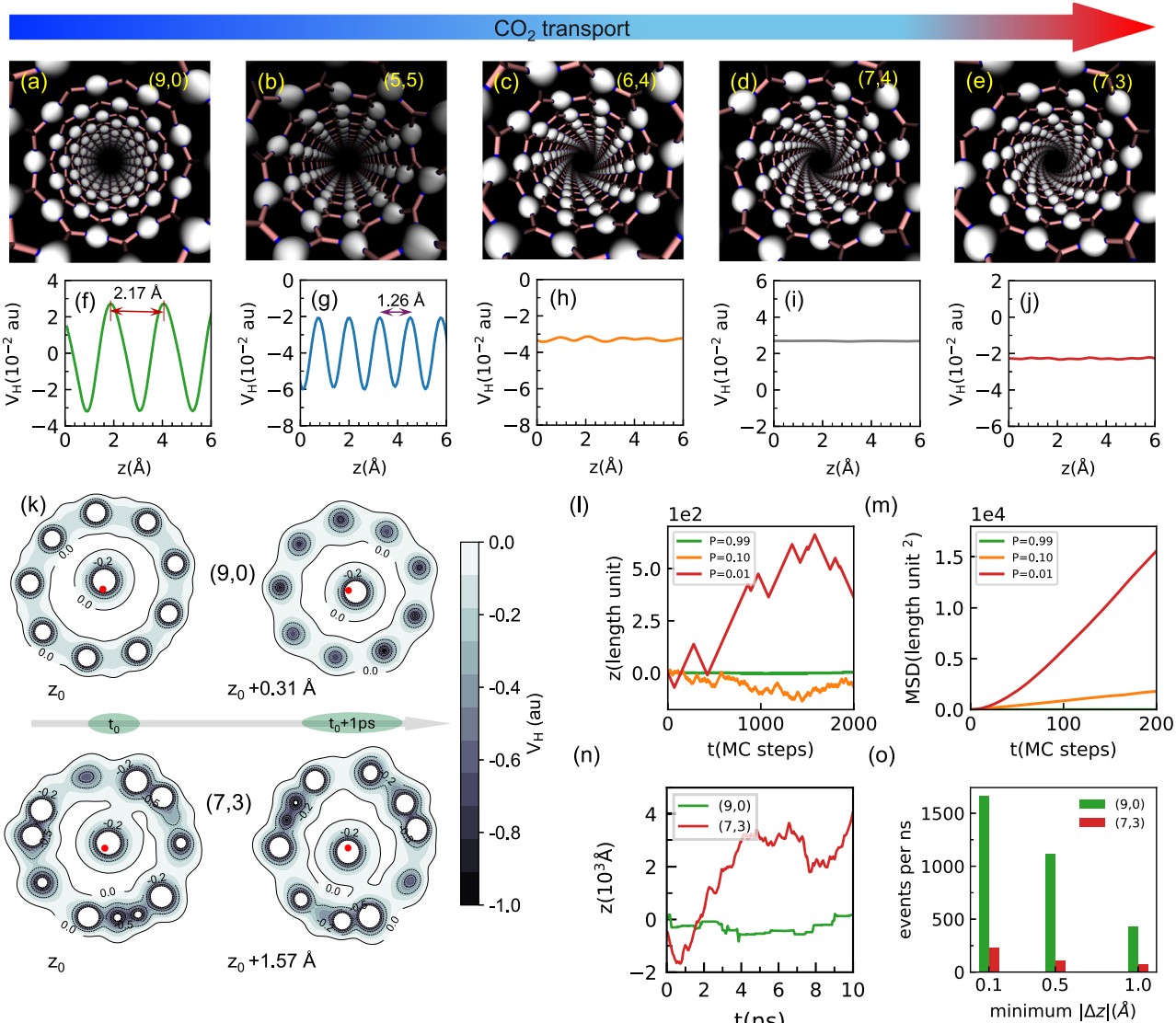

**Fig. 3 | Transport of CO$_2$ is connected to electronic properties of hBNNTs.**
**a–e** Visualization of electron clouds (density, iso-surface value = 0.3 au); **f-j** corresponding xy-average electrostatic potential VH (within 3.0 Å of the tube center, see S.1.3 for more details)along the tube direction; **k** cross-section of VH at the position of atom C (of CO$_2$) at 2 representative snapshots (1 ps apart) for the (9,0) and (7,3) systems, $t_0$ denotes the position of C at a time $t_0$ and the red dot denotes the tube's center; **l** the position of the walker in 1-dimensional MC simulations versus time (number of MC steps) with P being the probability of changing direction (left vs right); and **m** the corresponding MSD versus time (averaged from 1000 independent runs); **n** position of CO$_2$ in the tube vs time from MD; and **o** number of events that CO$_2$ reverses its moving direction (along the z direction), per ns, after travelling a minimum distance |Δz|.

## Molecular diffusion in selected tubes

Having identified tubes with large binding energies, we focused on probing CO$_2$ diffusion in tubes with similar radii, but different orientations of the hexagonal patterns. We selected one armchair (5,5), one zigzag (9,0), and three chiral tubes [(6,4), (7,3), and (7,4)] with differing degrees of twist and 3.45, 3.58, 3.47, 3.54, and 3.84 Å radii, respectively (the larger (7,4) tube was selected for comparison). These differences result in varied fundamental electronic properties, such as the distribution of electron clouds (Fig. 3a–e) or the xy-averaged electrostatic potentials inside the tubes (Figs. 3f–j and S2). Additionally, the selected tubes exhibit the highest N$_2$ adsorption capacities among those investigated (Fig. S3). Separations of CO$_2$ and N$_2$ using on hBNNTs will be discussed later.

MLIPMD simulations were subsequently conducted to determine the diffusivity of CO$_2$. We calculated the self-diffusion coefficient along the tube axis direction ($D_{self}$) using totally 10 ns statistics for each system, see "Methods" and Fig. S4. $D_{self}$ of CO$_2$ in the hBNNTs, Table 1, shows the tubes in increasing order of diffusivity: (9,0)

<< (5,5) < (6,4) ≤ (7,4) << (7,3), indicating that $D_{self}$ does not necessarily anticorrelate with the binding energy, see Fig. S5. It should be noted that the calculated $D_{self}$ from these simulations are for CO$_2$ with a flexible (slightly bent, for example, 174.1° in (7,3) on average) geometry that in part arises from interactions with the nitrogen orbitals inside the tubes. If CO$_2$ was constrained in a rigid linear geometry in a (7,3) tube, $D_{self}$ would drop by more than 2 times. This is consistent with previous studies that have shown faster diffusion for distorted, or "flexible" CO$_2$ molecules over linear, rigid ones[12]. Under confinement, deviations from linearity can reduce steric repulsion with the pore wall and enhance reorientational/librational motion, thereby strengthening translation-rotation coupling and facilitating transport. It is necessary to point out that the intrinsic bending mode of CO$_2$ lies at approximately 667 cm$^{-1}$ and remains largely quantum mechanical at room temperature, with its amplitude dominated by the ground-state wavefunction. Accordingly, numerical similarity between classical and quantum vibrational amplitudes is not interpreted as evidence that the bending dynamics are accurately described classically. In a confined

**Table 1 | Calculated self-diffusivity ($D_{self}$), Knudsen diffusivity ($D_K$), and selectivity in chiral and non-chiral nanotubes**

| Tube | Chiral | Tube diameter(Å) | CO₂ | | | N₂ | | | CO₂/N₂ |
|---|---|---|---|---|---|---|---|---|---|
| | | | $D_{self}$ (10⁻⁹ m²/s) | $D_K$(10⁻⁹ m²/s) | $D_{self}/D_K$ | $D_{self}$ (10⁻⁹ m²/s) | $D_K$ (10⁻⁹ m²/s) | $D_{self}/D_K$ | $D_{self}$ |
| (9, 0) | No | 7.16 | 271 (± 128) | 90 | 3 | 1017 (± 135) | 113 | 9 | 0.27 |
| (5, 5) | No | 6.90 | 2814 (± 299) | 87 | 32 | 2077 (± 348) | 109 | 19 | 1.35 |
| (6, 4) | Yes | 6.94 | 3252 (± 256) | 88 | 37 | 2402 (± 326) | 110 | 22 | 1.35 |
| (7, 4) | Yes | 7.68 | 3266 (± 268) | 97 | 34 | 1666 (± 203) | 122 | 14 | 1.96 |
| (7, 3) | Yes | 7.08 | 5521 (± 305) | 89 | 62 | 1625 (± 228) | 112 | 15 | 3.40 |
| (7, 3)_rigid | Yes | 7.08 | 2443 (± 151) | 89 | 28 | | | | |

In the case of (7,3)_rigid, CO₂ was constrained in a rigid linear geometry.

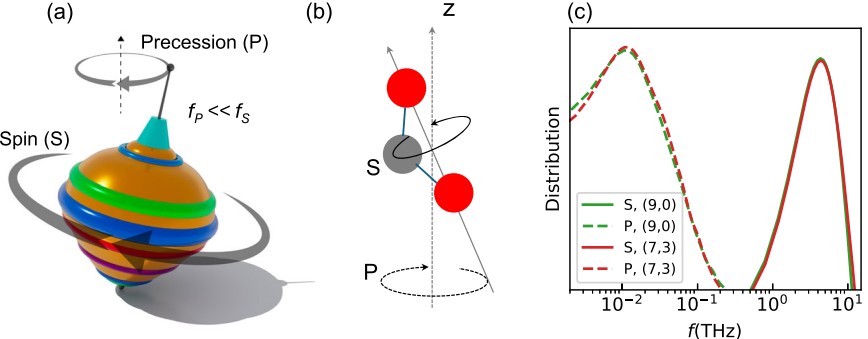

**Fig. 4 | Precession and Rotational Dynamics of CO₂ in hBNNTs.** a Precession of a spinning top; b scheme of two different rotations of CO₂, with C in grey and O in red; c frequency of rotation of CO₂ in hBNNTs.

pore, however, symmetry breaking leads to mixing between translational, rotational, and vibrational degrees of freedom, giving rise to low-frequency, thermally populated motions with a partial bending character. These nanoconfinement-induced modes are sensitive to the external potential and are well described classically. They can indirectly modulate the effective bending geometry and orientation of CO₂ without requiring direct classical treatment of the high-frequency quantum bending vibration.

Interestingly, there was no clear correlation between $r$ and $D_{Self}$, contradicting the notion that diffusion is faster in smaller diameter tubes[13,14]. The (5,5) and (6,4) tubes have very similar radii (0.6% difference), though the latter exhibits a ~1.2-fold enhancement in transport efficiency. Similarly, (9,0) has a 1.1% larger radius than (7,3); however, the CO₂ diffusion in (9,0) is 20 times slower. At first glance, the significantly slower diffusion of CO₂ in (9,0) is aligned with distinct peaks in the radial distribution functions of the C-B or C-N atom pairs (Fig. S6), different from the single peak feature of other studied systems. Nevertheless, the faster diffusion in the chiral tubes (particularly, in (7,3)) and lower diffusion in non-chiral tubes are attributed to various factors.

Of these factors, electrostatic potential is particularly important in our system due to the nature of the CO₂-hBN interactions (Van der Waals), with the Pauli repulsion between electron-rich atoms (O of CO₂ and N of hBN) heavily emphasized. The O atoms of CO₂ are likely to be at any locations inside the tube where repulsion is minimized. As such, the "localization" of CO₂ along the z-direction hinders the molecule's transport. Figure S7a–e shows the potential of mean force (PMF) with the z-component of the O-N distance as the reaction coordinate. The O atoms, and thus the CO₂ molecule, clearly favor particular locations in (9,0): the distance between two such locations along the z-direction is the same as the distance between two peaks of the electrostatic potential (Figs. S7a and 3f). To a lesser extent, CO₂ shows the same behavior in (5,5) (Figs. S7b and 3g). In the chiral tubes, the distribution is mostly even, aligning with the flat electrostatic potentials (Figs. S7c–e and 3h–j). The electrostatic maps in Fig. 3k demonstrate

the rotation of CO₂ about the tube axis. They also show high and low symmetry of the tube's electrostatic potential within a plane for (9,0) and (7,3), respectively. In contrast, the (average) potential along the z direction, which is subject to rotation, varies more significantly in (9,0) than in (7,3), which is consistent with the xy-average potentials shown in Fig. 3f, j). The electron-rich O atoms of CO₂ can experience stronger external local fields in non-chiral tubes than in chiral tubes due to the roughness of the electrostatic potential along the z-direction, Fig. 3f–j, decreasing CO₂ diffusivity through the non-chiral tubes.

How the roughness of the electrostatic potential hinders the diffusion of CO₂ along the z-direction of tubes can be further interpreted through Monte Carlo (MC) simulations of a 1-dimentionsal walker. Figure 3l–m demonstrates how the walker's (CO₂'s) mobility depends on the probability of reversing its moving direction. Higher reversal probabilities constrain the walker to oscillate around its initial position, yielding minimal net displacement. As the probability decreases, directional persistence increases, allowing the walker to move progressively farther from its origin. This transition forward, i.e., more extended diffusion, is captured by the corresponding rise in MSD, indicating that lower probabilities promote enhanced diffusivity. The MC results mirror the diffusion of CO₂ in the (9,0) and (7,3) tubes. Figure 3n-o shows that CO₂ moves much further in (7,3) than in (9,0) as time elapses and that CO₂ reverses its direction after travelling a given minimum distance much more frequently in (9,0) than in (7,3). The motion of CO₂ within the tube, including rotational behavior, may directly impact the collisions between the molecule and the walls.

**Precession-like behavior of CO₂**

Consistent with our initial hypothesis for this work, the CO₂ molecules were found to precess inside the tubes. Figure 4a shows the precession of a spinning top, with its two different co-occurring rotations: (1) spin, or the fast rotation about its own axis (S), and (2) the slow rotation of the spin axis about an external axis (P). Linear CO₂ is less diffusive than flexible CO₂, suggesting that molecular rotation may play an important role in this process. In this section, we analyze the rotation of CO₂

(specifically the C atom) about its own OO axis, facilitated by its bent geometry (Fig. 4b), and the rotation of the OO axis about the tube axis direction ($\bar{z}$), facilitated by the tilting of OO with respect to ($\bar{z}$) (Fig. S8). Here we focus on the two extreme cases: (9,0) and (7,3) tubes.

Movies S1–4 highlights both the slow, conics-like motion of the OO axis about the tube direction and the fast rotation of the molecular plane. The calculated S and P rotation frequency spectra (see Section S1.5 for additional details) are shown in Fig. 4c. In the two tubes, what we define to be a precession frequency is much lower (about two orders of magnitude) than the spin frequency, making it similar to the extremely slow precession trend of well-known objects such as Earth. To further understand the precession-like behavior of $CO_2$, we calculated the torque of a rigid, bent molecule in the two tubes (Fig. S9). While this artificial geometry is not physically accurate, it can be useful for interpreting the precession and the role of molecule-tube interactions in the absence of internal degrees of freedom. The z component of the torque ($\tau_z$), which makes the molecule rotate about the z direction, is higher in the (7,3) than in (9,0)−consistent with the trend of $f_P$ in these tubes. While the precession in (9,0) may not help $CO_2$ reduce collisions with the tube wall, Fig. 3k, the asymmetry of the potential may help in the (7,3) tube. Note that there are several factors dictating the unique transport of $CO_2$, we are pursuing further research into the underlying physics of the observed phenomena and will present it in a follow-on publication.

## Overcoming Knudsen diffusion and comparison with $N_2$ transport

Better understanding the effects of the topology of the interior of the chiral hBNNTs requires comparison with conventional diffusion mechanisms. We thus consider the Knudsen mechanism for both $CO_2$ and $N_2$ as a benchmark to determine if the precession-like motion has a significant effect on diffusion. Knudsen diffusion is active in our chosen system as the pore diameters (< 8 Å) of the studied hBNNTs are much smaller than the mean free path of both $CO_2$ and $N_2$ gas molecules and the densities of the gases are low, meaning that the gas molecules will collide with the pore walls more frequently than with other molecules[15]. The Knudsen diffusivity $D_K$ was calculated as

$$D_K = \frac{d}{3}\sqrt{\frac{8RT}{\pi M}} \qquad (1)$$

where $d$ is the pore diameter (m), $R$ is the gas constant (J/(mol K)), $T$ is the temperature (K), and $M$ is the molar mass (kg/mol)[16]. The $D_K$ of $CO_2$ and $N_2$ in different nanotubes were calculated and compared with the $D_{self}$ (shown in Fig. 3) in Table 1.

In the (9,0) tube, the $D_{self}$ of $CO_2$ is closer to the $D_K$. Distinctly different from this, in the chiral hBNNTs, the $D_{self}$ of $CO_2$ is much larger than the $D_K$, indicating a deviation from conventional diffusion mechanisms at these scales. The large $D_{self}/D_K$ ratio is particularly encouraging as the $CO_2/N_2$ Knudsen selectivity of 0.8 indicates that $CO_2$ being larger should exhibit inherently slower diffusion compared to $N_2$. Note that the $CO_2$ $D_{self}/D_K$ ratio in (5,5) is slightly reduced compared to (6,4) or (7,4), potentially due to the hyperloop effects discussed later.

Knudsen diffusion depends primarily on the pore size and the molecular mass (Eq. (1)), ignoring the rotational/vibrational degrees of freedom of $CO_2$ and interactions with the tubes. The amplification of self-diffusion over Knudsen diffusion, indicative of enhanced $CO_2$ diffusion in the chiral tubes, may be attributed to rotational degrees of freedom minimizing molecular collisions with the tube walls (see Supplementary Movies S1–4). The enhanced diffusivity of flexible $CO_2$ compared to linear $CO_2$, Table 1, demonstrates that the motion of a slightly bent $CO_2$ about its O−O axis (impossible in the rigid, linear geometry) significantly contributes to the molecule's ability to precess through the tubes, minimizing collisions and directional changes,

resulting in enhanced transport. In the non-chiral tubes, $CO_2$ still precesses, but experiences rougher free energy surfaces. This difference is reflected in the PMF and electrostatic potentials, Fig. S7 and Fig. 3f–j, resulting in more regular directional reversals and slower diffusion.

The relatively high $CO_2$ $D_{self}/D_K$ in non-chiral (5,5) can be linked to a reported "hyperloop" effect, which has shown that molecules confined and aligned within a 1-dimensional nano channel exhibit greatly enhanced diffusion[14]. Since this effect emerges when the molecular axis aligns with the nanochannel axis, we monitored the orientation of $CO_2$ within the tubes. The degree of alignment between $CO_2$ and each tube is quantified by α, defined as the angle between the O−O molecular axis and the tube axis (Fig. S9); smaller α values indicate greater alignment with the channel. The average value (in °) of α is 11.6, 9.0, 9.6, 17.9, 10.7, and the full width at half maximum (in °) of its distribution is 14.7, 11.3, 12.2, 22.0, and 13.5, in the case of (9,0), (5,5), (6,4), (7,4), and (7,3) tubes, respectively. In comparison to $CO_2$ in chiral (6,4), $CO_2$ in non-chiral (5,5) exhibits slightly more enhanced alignment and smaller variation in O−O direction, suggesting a lower degree of tumbling and a higher degree of hyperloop enhancement. This would counterbalance the stronger collisions between the molecules and nitrogen electron clouds in (5,5), leading to a similar $D_{self}/D_K$ ratio to (6,4), Table 1. $N_2$, however, exhibits very different behavior. In all tubes, $N_2$ undergoes tumbling during diffusion, retarding transport. In smaller tubes (5,5) and (6,4), $N_2$ exhibits higher diffusivity due to less frequent tumbling.

## Influence of the mechanics of tube geometry and wall interactions

To further our understanding of chirality-enhanced molecular transport for different molecules, we employed mathematical modeling that incorporates molecule-wall interactions that vary with molecular properties. As Knudsen diffusion depends primarily on the tube diameter and molecular mass (Eq. (1)), it neglects any mechanical interactions between the molecules and the tube walls. As the nanotube diameter approaches the molecular scale (~3.3 Å for $CO_2$), wall roughness and atomic-scale forces become significant, making the point-like particles approximation for the gas molecules invalid. In this scenario, the molecule-wall interactions play a more prominent role.

Figure 5 provides a qualitative description of the effect of precession on the molecule-wall interaction for different gas molecules where, $E$, $v$, and $t$ represent the circumferential Young's modulus, Poisson ratio, and tube thickness. At present, the causality between the molecule-wall interactions and precession-like molecular motion is unclear, but competition between thermal energy and molecular precession could lead to different modes of molecule-wall interactions and a different diffusion mechanism. Regardless of the specific underlying phenomena, we posit that the motion of $CO_2$ molecules is coupled to interactions with the wall along the chiral index, while that of $N_2$ is likely to be negligible.

For diatomic $N_2$ molecules, there is no molecular precession imparted by the tube wall. Thermal energy is dominant, leading to a more uniform molecule-wall interaction mode due to the random nature of thermal noise. The corresponding wall deformation $w$ is relatively uniform and small along the tube circumference. To compare the two deformations, we can calculate the uniform deformation $w$ subject to a uniform load $p$:[17]

$$w = \frac{p}{2E}[3(1-v^2)]^{\frac{1}{4}}\left(\frac{R}{t}\right)^{\frac{3}{2}} \qquad (2)$$

where $R$ and $t$ are the tube size and thickness.

At the other extreme, the transport mechanism imparted by wall interactions along the chiral index with $CO_2$ molecules could be strong enough to dominate over thermal energy. In this scenario, the coupled

**Fig. 5 | Schematic plot of different modes of molecule-wall interaction for $N_2$ and $CO_2$. a** Thermal energy is dominant for $N_2$ molecules. As diatomic $N_2$ does not exhibit molecular precession, the random nature of thermal noise tends to lead to a uniform molecule-wall interaction and deformation w; **b** an intermediate case between the two extremes; **c** molecular precession dominates over thermal energy for $CO_2$ interacting with nanotubes with strong chirality. The molecule-wall interaction tends to be less uniform and more concentrated, leading to a larger, localized deformation $w_2$ due to the effect of stress concentration.

molecule-wall interactions would be more concentrated and the corresponding wall deformation $w_2$ would localize and become much larger than the uniform deformation $w$ due to the effect of stress concentration. For comparison, we also calculate the localized deformation $w_2$ under concentrated load $F$[18]

$$w_2 = \alpha \frac{3\sqrt{2}(1-\nu^2)}{\pi} \frac{F}{Et} \left(\frac{R}{t}\right)^2 \qquad (3)$$

where $\alpha$ is a numerical factor of order unity. For a fair comparison, the two loads should be related as $F = 2pR$, as a result of requiring the same total molecule-wall interaction. With the Poisson ratio $\nu = 0.16261$[19], the rough estimate of the ratio between the two deformations now becomes: $w_2/w \approx 4(R/t)^{3/2}$, dependent on the ratio $R/t$. For single-wall hBNNTs, the size of the nanotube is comparable to the thickness[20] and the maximum ratio of localized to uniform deformation is around 4. This much larger localized deformation $w_2$ may contribute to the faster gas transport of $CO_2$ in chiral tubes. In contrast, smaller and uniform deformation may hinder the $N_2$ transport. This is directly analogous to driving a screw into wood, where the sharp threads concentrate stress and cause larger localized deformation, displacing material to enable fast forward motion along the axis.

The same mechanism is unlikely to operate for $N_2$ as it is a linear diatomic molecule that cannot bend and undergo precession-like motion like $CO_2$. Therefore, thermal energy always dominates for $N_2$ molecules, leading to uniform molecule-wall interactions. Thus, there is no localized deformation in chiral tubes for $N_2$ molecules, consistent with calculated diffusion rates for $N_2$ being slower than $CO_2$ as described above. While probing the specific underlying physical phenomena lies beyond the scope of this work, follow on studies of the physics and mechanics of the molecule-wall coupled interactions will shed light onto the observed enhanced diffusion.

### Studying diffusion for membrane applications

We assessed the potential performance of a hypothetical $CO_2$-separating membrane (most dense packing of an array of aligned cylindrical nanotube) with the selectivity and direction-specific diffusion found in the chiral BNNTs. It has been well documented that there is often a tradeoff between permeability and selectivity for polymer membranes. The relationships for many gas pairs were summarized into a series of plots known as the Robeson upper bound[21]. These upper-bound results were updated for polymer membranes in 2008 and are commonly used as a reference to understand the performance of gas separation membranes[22]. The transport of gases in a hypothetical membrane made of chiral (and non-chiral) hBNNT assembly can be

described using a solution-diffusion mechanism[23]

$$J_{CO2} = \left(\frac{D_{CO2}\varepsilon}{\tau}\right) A_m S_{CO2} \frac{\triangle f_{CO2}}{l} \qquad (4)$$

where the $J_{CO2}$ is the total flux of $CO_2$ across the membrane, $D_{CO_2}$ is the $CO_2$ diffusion coefficient in the nanotube, $\varepsilon$ is the membrane surface porosity, $\tau$ is the membrane pore tortuosity, $A_m$ is the membrane surface area, $S_{CO2}$ is the solubility coefficient of $CO_2$ defined as the ratio of $CO_2$ concentration in the nanotube to the $CO_2$ fugacity in the gas phase, $\Delta f_{CO2}$ is the fugacity difference between the bulk gas phases on either side of the membrane, and $l$ is the membrane thickness. Based on this mechanism, the permeability coefficient of the membrane to $CO_2$, $P_{CO2,}$ can be written as

$$P_{CO2} = S_{CO2} \times D_{eff, CO2} \qquad (5)$$

where $D_{eff,CO2}$ is the term in parenthesis in equation 4, called the effective diffusion (transport) coefficient of $CO_2$[24]. The effective diffusion coefficient was calculated based on the self-diffusion coefficients for the hBNNTs calculated in the molecular dynamics simulations detailed above (Table 1) multiplied by the porosity of the membrane and divided by the tortuosity of the pore in a membrane.

For an upper estimate of the effective diffusion coefficient, we assumed that a hypothetical membrane is made from a close-packed array of aligned cylindrical nanotubes, resulting in an average membrane porosity of 0.9069 and a pore tortuosity of 1. These self-diffusion coefficients of $CO_2$ in the BNNTs are comparable to those calculated for $CO_2$ diffusion in a (10,10) single-wall carbon nanotube at similar pressures with $CO_2$[25]. These self-diffusion coefficients are often used as a conservative approximation for the transport diffusion coefficient, reported for calculating the diffusion of argon (Ar) in carbon nanotubes[26].

For the (7,3) hBNNT, the effective diffusion coefficient $D_{eff,CO2} = 5521 \times \left(\frac{\varepsilon}{\tau}\right) = 5521 \times 0.9069 = 5007$ ($10^{-9}$ $m^2/s$) for the ideal membrane represents this upper limit. The porosity of some practical membranes range from 0.15 to 0.5[27]. A common relationship between the porosity and tortuosity of porous media is described by the Bruggeman equation[28]

$$\tau = \varepsilon^{-\frac{1}{2}} \qquad (6)$$

The corresponding tortuosity can be estimated from 2.6 to 1.4. A possible lower limit of the effective diffusion coefficient of $CO_2$ in a practical membrane is calculated to be $D_{eff,CO2} = 5521 \times \left(\frac{\varepsilon}{\tau}\right) = 5521 \times 0.0577 = 319$ ($10^{-9}$ $m^2/s$) using the same method shown above. Therefore, a possible lower limit of membrane permeance is about 1/15 that of the ideal membrane composed of the aligned array of nanotubes.

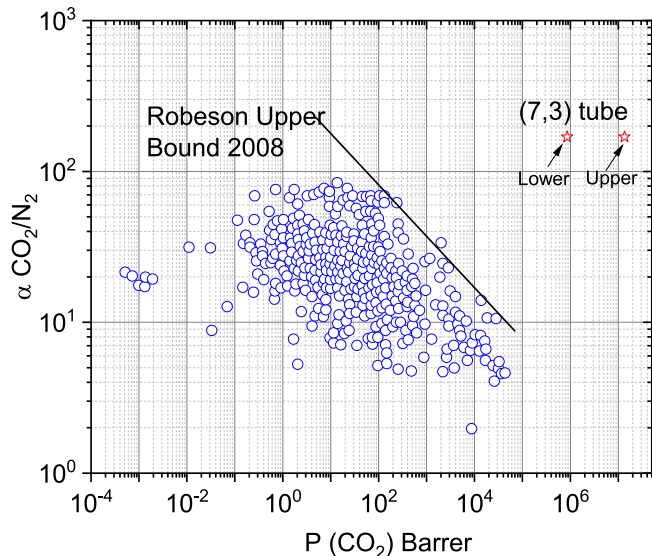

**Fig. 6 | The calculated $CO_2$ permeability coefficient in the 2008 Robeson upper bound relationship, data taken from ref. 22.** The star symbols define the possible lower and upper limits of the calculated permeability coefficients for the (7,3) hBNNTs studied here.

The $CO_2$ solubility coefficients for the hBNNTs were calculated by estimating the change in the Gibbs free energy from the bulk gas phase to inside the membrane pore. For simplicity, the calculation was done at room temperature and 1-atm pressure, where $\triangle G = \triangle G^0$ (see section S1): −2.0 kJ/mol (in (7,3) tube) for $CO_2$ and +7.7 kJ/mol (7,3) for $N_2$. From this, the above-defined solubility coefficient was calculated using the relationship between Henry's law constant and the change of standard chemical potential (equivalent to Gibbs free energy per mole)[29]

$$S = \frac{1}{RT}\exp\left(-\frac{\triangle G_0}{RT}\right) \qquad (7)$$

$$\triangle G = \triangle G_0 - RT\ln(f/f_0) \qquad (8)$$

where $\triangle G$ is the Gibbs free energy at fugacity of $f$, and $f_0$ is the fugacity at standard condition 1 atm, $R$ is the gas constant and $T$ is the temperature. Thus, the pure-component solubility $CO_2/N_2$ selectivity for (7,3) hBNNT is approximately 50, and the pure-component diffusion selectivity is 3.40 for $CO_2/N_2$. When combined, the overall pure-component permeability-selectivity for $CO_2/N_2$ in (7,3) hBNNT is estimated to be 170. This high permeability-selectivity suggests that hBNNT can be used for $CO_2/N_2$ separations. For the (7,3) hBNNT, the $CO_2$ solubility coefficient was estimated to be $9.17 \times 10^{-2}$ mol/L-atm while the $N_2$ solubility coefficient was estimated to be $1.83 \times 10^{-3}$ mol/L-atm. The permeance of the ideal aligned nanotube membrane can be calculated by multiplying the effective diffusion coefficient ($5007 \times 10^{-9}$ m²/s) and solubility coefficient ($9.17 \times 10^{-2}$ mol/L-atm). After converting the unit of permeability to a common unit, Barrer ($10^{-10}$ cm³(STP) cm² cm⁻³ cmHg⁻¹ s⁻¹), the permeability of the ideal membrane was estimated to be about $1.35 \times 10^7$ Barrer, considered as the upper limit. A potential lower limit of the permeance, as previously mentioned, was estimated to be about $8.61 \times 10^5$ Barrer.

Combined, the permeability-selectivity and upper and lower limits of the permeability coefficients for membranes made with the upper and lower limits of (7,3) hBNNTs were added to the 2008 Robeson upper bound plot of $CO_2/N_2$ separation (Fig. 6). These hypothetical membranes made of hBNNTs have the potential to break the 2008 upper bound for $CO_2/N_2$ separation. Even when the nanotubes (7,3) were included in a practical polymer matrix with the

porosity and tortuosity described above, the permeance and the $CO_2/N_2$ ratio of that membrane still would make it above the upper bound. Our results suggest that direction-specific diffusion could be a property that enables the design of next-generation gas separation membranes with *both* high gas permeability and selectivity. We are currently extending this work to include the concentration dependence of diffusivity, solubility, and permeability, as well as gas mixtures of $CO_2$ and $N_2$ instead of $CO_2$ and $N_2$ as pure components. These calculations will be useful to help guide potential separation applications.

### Expansion outside of separations of $CO_2$
This mechanism of diffusion may not be unique to $CO_2$ and may have been undetected for other molecules under nanoconfinement in chiral nanotubes. The literature includes a wealth of papers detailing the unexpectedly high diffusion and transport of water through some small-diameter CNTs. Holt et al. reported water flow rates that exceeded predictions from the Knudsen diffusion model by more than an order of magnitude and flow that exceeded values calculated from continuum hydrodynamics models by more than three orders of magnitude[30]. Tunuguntla et al. discovered highly accelerated water flow in short (~10-nm) fragments of CNTs embedded in lipid bilayer membranes[31]. While not specifically mentioned, the fast diffusion was observed in chiral CNTs. The modeled water molecules appear to be aligned in a linear fashion with a discernible rotation that matches the chiral indices of the CNT. Cambré et al. showed experimental evidence of water filling a chiral CNT with the molecules aligning single file[32], moving with a periodicity that appears to map to the CNT's chiral indices. These two studies suggest that the influence of chirality could impart a previously unconsidered means of diffusion for water (or clusters), similar to our findings on $CO_2$. Attempts to find more references about the effects of chiral tubes on water transport were limited as we found no mentions of the chiral indices of simulated CNTs, only diameters. We posit that a more thorough reassessment of the chirality of CNTs in prior work could elucidate whether undetected molecular precession may have influenced the rapid diffusion of water. If true, this finding may shed more light on why some CNTs rapidly move water while others do not. As such, we are currently initiating simulations to study the transport of water and other molecules in chiral nanotubes.

### Managing expectations: limits of experimental approaches
While this work has identified a molecular-level phenomenon with significant promise, we would be remiss if we did not highlight that these findings are from a computational study and not are not yet experimentally validated. A wide range of hBNNTs have already been synthesized, isolated, and characterized. Given sufficient quantities of pure hBNNTs, high-pressure magic angle spinning nuclear magnetic resonance, more specifically diffusion ordered spectroscopy, could measure diffusion coefficients of ¹³C-enriched $CO_2$ condensed inside the tubes. This approach has been used to study diffusion coefficients of $CH_4$[33,34], $CO_2$[33], and $H_2O$[33] in nanoporous media. Assessing the chirality of $CO_2$ diffusing through the tubes will be far more difficult to experimentally observe. We are in the process of designing such measurements by means of polarized light or polarized neutrons.

### Summary
This computational study presents evidence of a type of coupled transport for $CO_2$ confined inside chiral nanotubes that results in faster than Knudsen diffusion. hBNNTs distort $CO_2$ by ~5° through nanoconfinement and, when the chiral index provides a continuous twisted field, can coax $CO_2$ into molecular precession-like motion. Our calculations suggest that the (7,3) tubes are perfectly sized, achieving a high selectivity of $CO_2/N_2$. Chiral hBNNTs exhibit diffusion rates faster than non-chiral tubes of comparable or larger diameters. The (7,3) hBNNT

appears optimal, enabling $CO_2$ to diffuse faster than $N_2$ with a $CO_2/N_2$ diffusion selectivity of 2.06. The predicted performance of a hypothetical sheet membrane prepared with aligned chiral (7,3) hBBNTs shows a $CO_2/N_2$ permselectivity of 170 and a $CO_2$ permeance of $1.35 \times 10^7$ Barrer, readily surpassing the Robeson upper bound for $CO_2/N_2$ separation.

If the selectivity and diffusion can be experimentally validated, this molecular-level phenomenon could be further refined or exploited to design more advanced sorbents that could entail amplified diffusion by means of near-perfect direction-specific non-Knudsen diffusion of adsorbates down their axis. It is likely that molecular precession could be imparted by chiral carbon nanotubes or other porous materials like metal-organic frameworks, covalent-organic frameworks, and zeolites. We also assume that this phenomenon is not unique to $CO_2$ and that other molecules, like water, could achieve directional-specific diffusion down the axis of an intelligently designed sorbent.

## Methods
### Ab initio molecular dynamics (AIMD)
AIMD simulations were conducted using the CP2K program[35]. We employed the PBE-D3 density functional[36,37] and the hybrid Gaussian–Plane wave basis set framework[38], in which the DZVP Gaussian basis functions[39] in conjunction with a plane wave cut-off of 450 Ry were used. The Goedecker-Teter-Hutter pseudopotentials[40] with valance states of B($2s^2$ $2p^1$), C($2s^2$ $2p^2$), N($2s^2$ $2p^3$), and O($2s^2$ $2p^4$), were adopted. In self-consistent calculations, only the Γ-point was considered. All molecular dynamics simulations were carried out within the NVT ensemble, in which the temperature was maintained at 300 K with the canonical sampling through velocity rescaling thermostat[41]. The time step was set at 1.0 fs. Each system was equilibrated for 10 ps, followed by a 70-ps production run. For simulations involving rigid $CO_2$, the three internal bond distances (C–O, O–C, and O–O) were constrained using the SHAKE algorithm[42], with a tolerance of $5 \times 10^{-5}$.

### Static density functional calculations for binding energies
The binding energy of a (confined) $CO_2$ or $N_2$ molecule (mol) in a hBNNT was calculated as

$$\Delta E = E(mol - hBNNT) - E(mol) - E(hBNNT) \quad (9)$$

in which the energy $E$ of each system was calculated at the density functional level provided above.

### Deep potential model training
We used the deep potential (DP) model[43], implemented in the DeePMD-kit[44], to train machine learning interatomic potentials. Very briefly, the total potential energy of a system is given by

$$E = \sum_i E_i = \sum_i N(D_i(\boldsymbol{R}_i)) \quad (10)$$

in which $E_i$ is the local atomic energy determined by atom i and its surrounding environment within a cutoff $R_c$, the symmetry-preserving descriptor $D_i$ is the feature matrix encoding the surrounding environment and is fed to a deep neural network $N$ which returns the energy $E_i$. $\boldsymbol{R}_i$ denotes the set coordinates of all atoms in the environment, $\boldsymbol{R}_i = \{\boldsymbol{r}_{ij} \equiv \boldsymbol{r}_i - \boldsymbol{r}_j\}$.

The network is trained through the minimization of the loss function

$$\mathscr{L} = p_E |\triangle E|^2 + \frac{p_f}{3N} \sum_i |\triangle F_i|^2 \quad (11)$$

where $\triangle E$ and $\triangle F$ are the deviation of the potential energy and atomic forces between the reference DFT and predicted data, respectively; and $p_E$ and $p_f$ are tunable prefactors.

We used a {20,40,80} embedding and {200,200,200} fitting network. The radial cutoff and the smooth cutoff were set at 6.5 and 4.5 Å, respectively. The prefactor $p_E$ was set to increase from 0.02 to 1 and $p_f$ was set to decrease from 1000 to 1.

From 70,000 AIMD frames, 56,000 frames were randomly chosen to create a training set, similarly, 7000 frames for a validation set and 7000 frames for a test set. The accuracy of trained potentials is provided in Table S1.

### Machine learning interatomic potential molecular dynamics (MLIPMD)
MLIPMD simulations were carried out using LAMMPS[45]. We used the NVT ensemble in which the temperature (300 K) was maintained using the Nose-Hoover thermostat[46]. The time step was set at 1 fs. Five independent simulations were conducted for each system, each comprising a 0.5 ns equilibration followed by a 2.0 ns production run with atomic coordinate recorded every 20 fs. This resulted in a total of 10 ns of production data per system for statistical analysis. In simulations with rigid $CO_2$, the "rigid/nvt molecule" command was applied to constrain intramolecular motion while maintaining a constant temperature ensemble.

### Diffusion coefficient
The self-diffusion coefficient of $CO_2$ along the tube direction z, $D_{self}$, was calculated from the mean square displacement ($MSD_z$)

$$D_{self} = \frac{1}{2} \lim_{t \to \infty} \frac{d}{dt} MSD_z(t) \quad (12)$$

$$MSD_z(t) = \left\langle |r_z(t) - r_z(0)|^2 \right\rangle \quad (13)$$

in which $r_z$ is the z component of the centroid coordinate of a $CO_2$ molecule and the <> notation indicates the ensemble average.

## Data availability
The MLIP files are available upon request. Source data for figures in the main text are provided with this paper. Structures (in the xyz format) of $CO_2$ and $N_2$ in the (7,3) and (9,0) hBNNTs are provided as files (Supplementary_XYZ_1_CO2_hBN73.xyz, Supplementary_X-YZ_2_CO2_hBN90.xyz, Supplementary_XYZ_3_N2_hBN73.xyz, Supplementary_XYZ_4_N2_hBN90.xyz). Source data are provided with this paper.

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

## Acknowledgements

This work was supported by the U.S. Department of Energy (DOE), Office of Science, Basic Energy Sciences, Chemical Sciences, Geosciences, and Biosciences Division, Harnessing Confinement Effects, Stimuli, and Reactive Intermediates in Separations, FWP 81462. (insert other projects describing work their grants supported here) Pacific Northwest National Laboratory (PNNL) is operated by Battelle for the U.S. DOE under contract DE-AC05-76RL01830. This research also used resources of the National Energy Research Scientific Computing Center (NERSC), a Department of Energy Office of Science User Facility using NERSC Award No. BES-ERCAP0031452. The authors thank Dr. Joseph Swisher for discussions on the calculated diffusion and Robeson analysis and Dr. Greg Schenter and Dr. Chris Mundy for fruitful physics discussions. The authors also thank Mr. Mike Perkins and Mr. Cortland Johnson for graphics design and support and Dr. Beth Mundy for technical editing.

## Author contributions

Nguyen and Heldebrant jointly conceived the study. Nguyen performed designed and performed the modeling studies, Heldebrant secured funding and managed the project. Liu and Bhown performed

calculations of membrane performance. Xu performed mathematical modeling. All authors wrote the paper.

## Competing interests

The authors declare no competing interests.
