## [Transparent Peer Review file · Nature Communications]

Direction-Specific Enhanced Diffusion of CO₂ in Chiral Hexagonal Boron Nitride Nanotubes

Corresponding Author: Dr David Heldebrant

Version 0:

Reviewer comments:

Reviewer #1

(Remarks to the Author)

This study examines an interesting approach to enhancing CO₂ permeability via chirality-induced spinning in hexagonal boron nitride nanotubes (hBNNTs). The effect of nanotube structure on CO₂ and N₂ diffusion was simulated. The idea is new, and the results are informative to the community. My other comments are shown below.

1. Table 1 shows CO₂ diffusion coefficient in various tubes but only one N₂ diffusion in (7,3). Can the N₂ diffusivity in other tubes be calculated?
2. Please justify the use of the sorption-diffusion model ($P = S \times D$). This model is often used for nonporous materials, where gas molecules need to be dissolved in the materials first. If gas diffusion follows Knudsen diffusion, then the sorption might not play a role.
3. It was widely postulated that gas diffusion in nanotubes may have a slipping boundary (as suggested by Holt's Science article, Ref. 29), and thus gas diffusion is much higher than Knudsen diffusion. Please comment if this could be true in the hBNNTs.

Reviewer #2

(Remarks to the Author)

This study investigates a novel diffusion mechanism in chiral hexagonal boron nitride nanotubes (hBNNTs), wherein the chirality of the nanotubes imparts a "rifling" effect that aligns and enhances CO₂ diffusion along the nanotube axis. Through first-principles simulations, the authors demonstrate that CO₂ diffusion in chiral hBNNTs can significantly surpass that in non-chiral nanotubes of similar diameters. The study also evaluates the feasibility of a membrane-based separation system leveraging this directional diffusion to improve CO₂/N₂ selectivity. While the concept is innovative, some inconsistencies in the data and diffusion trends raise questions regarding the fundamental transport mechanisms at play. Specifically, the diffusion enhancement and adsorption behavior require further scrutiny to establish their validity and consistency across different nanotube structures.

1. The analogy between CO₂ diffusion and the movement of a bullet is highly misleading. In ballistics, the orientation and rifling effects of a bullet influence the flow pattern of air surrounding it, thereby altering the frictional forces that are crucial for stability and accuracy. Friction is the dominant factor affecting a bullet's trajectory, making rifling essential. However, in gas diffusion at the molecular scale, there is no comparable frictional force acting on CO₂ molecules. The orientation and spin of CO₂ do not produce the same stabilizing or accelerating effects seen in bullets. Instead, the movement of CO₂ is dictated by adsorption-desorption dynamics, interactions with the nanotube walls, and random thermal motion. The bullet analogy is simply wrong.
2. The study asserts that chiral hBNNTs enable a "non-Knudsen" diffusion mechanism with enhanced directionality. However, Table 1 shows that for the non-chiral (9,0) nanotube, the self-diffusivity of CO₂ is lower than its Knudsen diffusivity ($D_{\text{self}}/D_K = 0.82$), whereas for chiral tubes, the ratio is higher. Given that Knudsen diffusion should generally dominate at these length scales, how do the authors justify the claim that the rifling-induced spin is the primary factor in enhancing CO₂ transport, rather than simple variations in adsorption strength or tube diameter effects?

3. The manuscript primarily focuses on CO₂ diffusion, with minimal discussion on N₂ diffusion behavior. The data suggest that N₂ diffusion is largely unaffected by the chiral characteristics of the nanotubes, yet no detailed explanation is provided. What is the underlying reason for this discrepancy? If chirality significantly influences CO₂ transport, why does it not induce a similar effect on N₂ diffusion? Is this due to differences in molecular interactions with the hBNNT surface, differences in quadrupole moment, or another factor?

4. The manuscript does not provide sufficient details on the adsorption behavior of CO₂ and N₂ within the nanotubes. If chirality significantly influences CO₂ diffusion, it is reasonable to expect a similar effect on CO₂ adsorption. Do the adsorption energies or molecular orientations of CO₂ inside chiral nanotubes differ from those in non-chiral nanotubes? Furthermore, the authors should clarify whether N₂ adsorption is similarly affected by chirality. If N₂ diffusion remains relatively unchanged, does this imply that the adsorption characteristics of N₂ are also independent of chirality?

5. The authors claim that the chiral nanotubes cause CO₂ diffusivity to deviate significantly from Knudsen diffusivity. However, the simulation results presented in Table 1 indicate that even non-chiral nanotubes, such as (5,5) and (9,0), exhibit diffusivities that differ notably from Knudsen predictions. This suggests that factors other than chirality are influencing diffusion behavior. Could the observed deviations be attributed to variations in pore size, surface roughness, or adsorption effects rather than the rifling mechanism?

6. The study presents adsorption energy trends indicating that the (7,3) nanotube has the strongest CO₂ binding energy (-57 kJ/mol). However, strong adsorption is typically associated with slower diffusion due to increased residence time. How do the authors reconcile this contradiction?

Reviewer #3

(Remarks to the Author)

The authors investigate the rifling transport of CO₂ using molecular dynamics, aiming to find applications in the field of gas separation. As highlighted by the authors, molecular transport across nanotubes has been a focal point of interest for uncovering novel mass transport phenomena in confined spaces. In this study, the authors delve into understanding the rifling transport of CO₂ within chiral hexagonal boron nitride nanotubes. Although the concept is highly interesting, the analysis remains superficial, and the statistics presented appear questionable. The authors are strongly encouraged to provide further details on both the systems utilized and the statistical methods applied to ensure the robustness and credibility of their findings. A comparison with current technologies is highly beneficial to engage a broader readership while carefully emphasizing both the limitations of the study and their future research plans.

Major Comments

1. The authors report a very interesting CO₂ transport phenomenon across chiral hexagonal boron nitride nanotubes. However, the chosen system and the presented statistics are elusive and unclear to the reader. A commonly employed strategy for simulating membrane-based processes involves using a pressurized feed gas that undergoes separation at a selective interface (in this case, the nanotube film). By recording the number of escaped molecules, it is possible to estimate gas permeance and selectivity. According to the methods section, there are serious concerns regarding the statistical reliability of the study, given the very short simulation times, which are limited to a maximum of 70 ps. Furthermore, the absence of a sufficiently large simulated system to provide robust permeation statistics significantly undermines the validity of the findings. Without strong statistical proof of the system, I cannot recommend its publication in Nature Communications.

2. The gas force fields discussion is missing in the method sections. It is particularly important to report both non-bonded and bonded interactions while CO₂ bond flexibility is at the heart of this work.

3. The potential of mean force (PMF) would be ideal to reinforce the claims of the manuscript. By sampling the free energy profile across the different nanotubes, one can assess precisely which nanotube would lead to a faster permeation rate. Authors are highly encouraged to run analysis through selective nanotube channels to reinforce the discussion about permeation.

Minor comments:

4. Figure 3 caption is quite confusing for the reader.

5. The manuscript would really benefit in clarity by name chiral and non-chiral CNT with distinctive names such as "chi" at the end of chiral CNT names. This would enable smoother discussions.

6. The manuscript would benefit from detailed method on how the electrostatic potential was computed.

Reviewer #4

(Remarks to the Author)

The authors have likely identified a significant and interesting phenomenon with publishable data. There are however many technical deficiencies in the manuscript that need to be addressed before the manuscript can be more carefully reviewed. The manuscript has some strong aspects including a compelling result and some good art work. It is also scientifically quite sloppy. The manuscript needs a careful makeover and re-review before it can be assessed properly.

1. An issue is that if the authors want to consider the vibrational distortion of the molecule as a critical parameter this cannot be obtained from purely classical dynamics. The nuclear dynamics i.e. vibrations are quantized and distortions would not be expected to be represented in this work using DFT MD. The chirality of the tubes, as is well demonstrated in the animations that are quite nice, defines a preferred direction of diffusion. It is not at all obvious to me that this phenomenon would not occur with rigid a rigid CO₂ model. The idea of the spinning molecule based on the distorted geometry is unconvincing as presented. Indeed, I am not sure at all that is what they are seeing. Checking to see if even an imperfect empirical MD simulation with rigid CO₂ molecules reproduces the phenomena could address several issues including the time scale concerns in modeling diffusivity. Also, a spinning motion would imply a large moment of inertia with respect to the small vibrational amplitudes and large energies I think.

2. It would be helpful if the authors explained the very different binding mechanism for (9,0) and all the other systems as shown in the SI radial distributions. The binding energies are comparable, but this is clearly quite distinct.

3. The authors state "CO₂ is very close to the wall in tubes with small radii, resulting in large Pauli repulsions and weakening the CO₂-hBNNT bonding." There is little evidence of this in Fig. 2. The confined tubes have strong comparable net binding energies and little evidence of weakened bonds. I am not sure what the point is here and it distracts from the basic point that confinement greatly enhances favorable dispersion interactions, especially compared to flat surfaces. It seems these strong interactions might be enhancing directional diffusion in the chiral tubes?

4. The authors follow up by stating "Having identified tubes with optimal radii and adsorption energies, we focused on probing CO₂ diffusion in tubes with near identical radii, but with different orientations of the hexagonal patterns." In what sense is this optimal? It is far from obvious that the strongest binding energy is best for diffusivity. The connection needs to be made to state this.

5. This is from the text:

Static density functional calculations for binding energies

The binding energy of a (confined) CO₂ molecule on a BNNT is calculated as

$$\Delta E = E(\text{CO}^{\text{in}} - \text{BNNT}) - E(\text{CO}^{\text{out}}) - E(\text{BNNT}),$$

in which the energy E of each system was calculated at the density functional level provided in S1.2.

This is from the SI:

S1.2. Static density functional calculations for binding energies

37

The binding energy of a (confined) X (X=CO₂, N₂) molecule a hBNNT was

38

calculated as

39

$$\Delta E(X) = E(\text{CO}_2 - \text{hBNNT}) - E(X) - E(\text{hBNNT}),$$

40

41

in which the energy E of each system was calculated at the density functional level

42

provided in Materials and Methods (main text).

So how was it calculated? This is sloppy.

6. A minor point but these are not really AIMD but rather use DFT methods. DFT MD or periodic electronic structure MD are preferred IMO.

7. The text uses free energy estimates that are given but not in any way justified in the SI. This needs to be addressed.

8. I am concerned about the MSD calculations. The authors need to show they are linear. They seem to compare to a more approximate method in the SI (Fig. S2,S3) but only show dynamics over the same short time range. It is not at all clear they are in the linear regime. I believe they are seeing a real phenomenon, but this is a high level journal and the work is not thorough or ostensibly carefully done.

9. The distribution of angles is given in the SI but this is not tied to any proof of a rifling effect in the text. As I said I am not convinced of this mechanism but if they want to claim this as the reason it needs to be justified.

10. The videos show a regular librational motion that in the diffusive (7,3) is apparently coupled to directional transport and the higher amplitude (9,0) seems to be normal diffusion. This coupling of motions does seem to be at least a part of the observed directional diffusion mechanism. Is this what the authors mean by hopping? The S6 figure does not show anything exceptional that correlates.

I could continue but the authors need to take some time to better present this very compelling and interesting work in a more carefully.

Version 1:

Reviewer comments:

Reviewer #1

(Remarks to the Author)

Through simulations, this work unveils an interesting transport mechanism for CO₂, i.e., direction-specific diffusion. With appropriate size of hexagonal boron nitride nanotubes, CO₂ permeability of 1.35×10^7 Barrer and CO₂/N₂ selectivity of 170 were achieved. This concept is novel. My other comments are shown below.

1. Knudsen diffusion was calculated inside the tubes. However, the tubes have sizes (0.6-0.8 nm) close to the CO₂ molecule (0.33 nm). Should other diffusion mechanisms be considered, such as blocking or surface diffusion? The blocking model has also been used for inorganic molecular sieve membranes.
2. The permeability was estimated based on a porosity of 0.9069, which seems to be very high. Is it possible to estimate a more real porosity based on other nanotube studies such as CNTs?
3. For the membrane calculation, what is the membrane thickness?

Reviewer #2

(Remarks to the Author)

The central concept of “molecular spin” and “rifling motion” of CO₂ in chiral hBNNTs is physically questionable and insufficiently supported. According to group theory, a linear centrosymmetric molecule such as CO₂ does not possess a genuine rotational degree of freedom about its molecular axis, and thus cannot sustain the type of axial spin the authors propose.

The “spin” observed in their simulations appears to arise only from slight bending distortions of the molecule under confinement, rather than from any well-defined angular momentum transfer. Furthermore, the analogy to rifled bullets is physically misleading, since diffusion in nanopores is dominated by Brownian dynamics and interaction potentials rather than torque-induced rotation. While confinement and chirality may indeed bias molecular orientation and diffusivity, the claim of a “nano-rifling” mechanism is very misleading, as it currently lacks experimental validation and risks overstating the physical plausibility.

I therefore cannot support publication of this work unless such terminology is removed.

Reviewer #3

(Remarks to the Author)

The authors have addressed my comments.

Reviewer #4

(Remarks to the Author)

I am happy overall with the response to all the reviewers and the thorough modifications. I will ask that they modify some language here (see below) and their response is actually wrong. As they have established very nicely, the chiral transport depends on the molecular distortion of the CO₂ molecule (the CO₂ bend freq, us 667 cm⁻¹ e.g.). These bends and such are quantum mechanical vibrations at multiples of thermal energy. Thus, their interaction, distortions, vibrational amplitudes etc. are not at all described in DFTMD or using empirical potentials. This is simple physics. What is established is that the distortion couples to the chiral PES and changes the dynamics. I see no reason this won't carry over to the true dynamics that is not easy to simulate but mostly for technical reasons. So I strongly suggest the authors briefly talk about this issue and maybe even give the ratio of the amplitudes for say the quantum harmonic bend to the classical result for the relevant temperature and that suffices with some logic as I outline here. It is an interesting result and I complement the authors for taking the reviewers seriously.

a. To the first point “An issue is that if the authors want to consider the vibrational distortion of the molecule as a critical parameter this cannot be obtained from purely classical dynamics. The nuclear dynamics i.e. vibrations are quantized and distortions would not be expected to be represented in this work using DFT MD.” It is true that the theoretical framework employed in this study does not capture nuclear quantum effects. However, these effects might be important in H-containing systems like water. Nevertheless, for studying transport of water in nanotubes in recent literature, methods that incorporate quantum nuclear effects like path integral molecular dynamics are rarely used, purely classical dynamics approaches are

widely adopted (see for example, <https://www.pnas.org/doi/epdf/10.1073/pnas.2211348119>). b. To the second point "The idea of the spinning molecule based on t

Version 2:

Reviewer comments:

Reviewer #1

(Remarks to the Author)

The authors have sufficiently addressed my prior comments, and the manuscript may be considered for publication in this premier journal.

Reviewer #4

(Remarks to the Author)

The authors have made progress addressing my main concern. Still, I think some appropriate language and or calculations need to address the issue in the manuscript.

The basic idea is that a purely ground state dominated quantum mechanical bending mode of a linear molecular couples to an external potential (provided by the material) in such a way that it bends driving a unidirectional diffusion during its dynamics. They have strong classical evidence as they turn up the frequency (rigid) and eliminate the behavior by reducing the coupling.

I think I understand the confusion. One might think because we could say optimize the molecular structure in the material and obtain forces and repeat and get a series of confirmations that all seem reasonably obtained. In the paper they use a finite temperature empirical potential, but similar considerations arise. I think this is why they keep wanting to ignore quantum mechanics and feel justified.

Consider though as the frequency of the mode increases the amplitude in QMs is not temperature dependent and only relatively weakly indirectly frequency dependent through the dependence of the ground state wavefunction of the potential. Further, this all depends on the coupling to the external potential. The linear molecular in the material will not have pure rotations or vibrations, the modes mix giving the vibrations rotational character that is what presumably causes the effect they see from this perspective. Still the molecule must bend to get the coupling required I believe and this mode is likely to be high frequency and quite quantum mechanical. Some of this might be sorted out by getting the normal modes of the system for a few conformations. As I said last round the classical effect is likely to be preserved even in the real world. But it is up to the authors to explain this in the manuscript in my opinion. It is a requirement to be in a good journal that the physics is well explored. It is a tricky situation and perhaps I am missing something.

As for the previous response:

"= 667 cm^{-1} , this ratio of amplitude is 0.76 if T=300 K. Thus, the classical approach, although underestimating the amplitude, fairly describes the bending of CO₂. This suggests that the bending (thus, rotation effects) is more likely to be driving this phenomenon. ." This is not how dynamics works coupling to a QM degree of freedom. With some effort the suggestion and analysis has value but it is not "fairly described" as a result of the amplitudes being similar. This is bad. You can't just ignore QM degrees of freedom that are centrally involved in the phenomena of interest because the codes you use don't handle them or neglects them. It may just be as I suggest that the geometry distortion induced by the material brings in to play the low frequency modes that move the system by a combination of rotation / translation of the molecule in the material system that can be treated classically and then the issue is mostly addressed.

Version 3:

Reviewer comments:

Reviewer #4

(Remarks to the Author)

I am ok with proceeding. I think the modes showing up in the classical approximate density of states is a strong argument. To be clear, they haven't shown that if they mapped the potential energy that the Born-Oppenheimer surfaces would result in low frequency distortions that can be described classically. There is still a nonzero chance that the effect seen classically is an artifact of classical mechanics where all modes are thermally populated. Nonetheless the low frequency spectrum is sufficiently suggestive and as the authors point out this is a aspirational phenomena that has value as such.

RESPONSE TO REVIEWERS' COMMENTS

Reviewer #1 (Remarks to the Author):

This study examines an interesting approach to enhancing CO₂ permeability via chirality-induced spinning in hexagonal boron nitride nanotubes (hBNNTs). The effect of nanotube structure on CO₂ and N₂ diffusion was simulated. The idea is new, and the results are informative to the community. My other comments are shown below.

We thank the reviewer for his/her assessment of our work and comments/suggestions.

1. Table 1 shows CO₂ diffusion coefficient in various tubes but only one N₂ diffusion in (7,3). Can the N₂ diffusivity in other tubes be calculated?

Response 1:

We thank the reviewer for this ask. We have further calculated the N₂ diffusivity all tubes, using molecular dynamics based on machine learning interatomic potentials. It appeared the N₂ diffusivity is lower than that of CO₂ in all tubes except for (9,0) - a nonchiral tube. We Discussion of new data and comparison with CO₂ have been implemented in the revised manuscript.

2. Please justify the use of the sorption-diffusion model ($P = S \times D$). This model is often used for nonporous materials, where gas molecules need to be dissolved in the materials first. If gas diffusion follows Knudsen diffusion, then the sorption might not play a role.

Response 2:

We thank the reviewer for the comments. There are two common models, pore flow model and solution-diffusion model, to describe the transport of molecules through membranes. The transport of CO₂ and N₂ molecules through the hypothetical membrane is certainly not controlled by size exclusion so we chose the solution-diffusion model for our calculation. In addition, it is proposed that these two models could be united to explain the results for membrane exhibit features common to both the pore flow model and solution-diffusion model (Hedge et al., Science. 2022, 377, 186-191. DOI: 10.1126/science.abm7192) Our molecular dynamics simulations indeed predict in most cases the gas diffusivity is higher than Knudsen diffusivity, so the dissolution/adsorption generally helps the transport of gas molecules.

3. It was widely postulated that gas diffusion in nanotubes may have a slipping boundary (as suggested by Holt's Science article, Ref. 29), and thus gas diffusion is much higher than Knudsen

diffusion. Please comment if this could be true in the hBNNTs.

Response 3:

Gas molecules interact with the wall of the pores and partition between the gas phase and the solid phase. The mechanism for gas diffusion can be Knudsen, Fickian, or something else. As we mentioned in the previous response, adsorption dictates the concentration of the gas molecules in the pore and can enhance gas diffusion through the membrane. We believe that if there is no strong “adsorption”, “hyperloop”, or “chirality” effect, there should be no significant difference between gas diffusion and Knudsen diffusion for gas penetration through a membrane (see Eqs. 5 and 6).

To further strengthen our understanding of CO₂ Knudsen diffusivity deviation from calculated diffusivity, we have added a section “*Mechanics aspects of nano-riofling and spin-enhanced transport*” to the revised manuscript:

“While the difference between the zigzag (9,0) tube and the other tubes is remarkable, the difference between the armchair (5,5) and chiral tubes is less pronounced. To further our understanding of the chirality-enhanced molecular transport, we employed mathematical modeling incorporating mechanical effects which vary with the twist angle relative to the armchair tube (see section S1). The Knudsen diffusion depends on the tube diameter and molecular mass (Equation (1)), neglecting any mechanical interactions between the molecules and the tube walls. As the nanotube diameter approaches the molecular scale ($\sim 3.3 \text{ \AA}$ for CO₂), the wall roughness and atomic-scale forces become significant, making the point-like particles approximation for the gas molecules invalid. In this scenario, the molecule-wall interaction plays a more prominent role.

Figure 6. Schematic plot of different modes of molecule-wall interaction for hBNNs of different chirality, represented by the twist angle. a) Thermal energy is dominant over the molecular spin in nonchiral nanotubes. The molecule-wall interaction tends to be uniform due to the random nature of thermal noise, resulting in a uniform deformation w ; b) Intermediate case between the two extremes; c) Molecular spin is dominant over the thermal energy in nanotubes with strong chirality. The molecule-wall interaction tends to be less uniform and more concentrated, leading to a larger, localized deformation w_2 due to the effect of stress concentration.

Figure 4 provides a qualitative description of the effect of chirality on the molecule-wall interaction. Here, E , ν , p , and t represent the circumferential Young's modulus, the Poisson ratio, and the tube thickness. The spin of bent CO_2 molecules, imparted by the nano-rifling phenomenon, enhances the stability of the motion of molecules through the gyroscopic effect, creates the angular momentum, stabilizes the molecule's trajectory, and reduces the thermal tumbling. Particularly, the competition between the thermal energy and the molecular spin could lead to different modes of molecule-wall interactions.

For an armchair nanotube (5,5), which is symmetric and nonchiral with twist angle θ , there is little spin imparted by the tube wall, such that thermal energy is dominant, leading to a more uniform molecule-wall interaction mode due to the random nature of thermal noise. The corresponding wall deformation w is relatively uniform and small along the tube circumference. To compare the two deformations, we could calculate the uniform deformation w subject to a uniform load p :¹⁹

$$w = \frac{P}{2E} \left[3(1-\nu^2) \right]^{\frac{1}{4}} \left(\frac{R}{t} \right)^{\frac{3}{2}}, (7)$$

where R and t are the tube size and thickness.

With increasing chirality, on the other end of the extreme, the spin imparted by the chiral nanotubes could be strong enough and dominant over the thermal energy. In this scenario, molecule motion is stabilized by molecular spin, and the molecule-wall interaction could be less uniform and more concentrated. The corresponding wall deformation w_2 is also localized and

could be much larger than the uniform deformation w due to the effect of stress concentration. For comparison, we also calculate the localized deformation w_2 under concentrated load F :²⁰

$$w_2 = \alpha \frac{3\sqrt{2}(1-\nu^2)}{\pi} \frac{F}{Et} \left(\frac{R}{t}\right)^2, \quad (8)$$

where α is a numerical factor of order unity. For a fair comparison, the two loads should be related as $F = 2pR$, as a result of requiring the same total molecule-wall interaction. With the Poisson ratio $\nu=0.16261$,²¹ the rough estimate of the ratio between the two deformations now becomes: $w_2/w \approx 4(R/t)^{3/2}$, dependent on the ratio R/t . For single wall hBNNTs, the size of the nanotube is comparable to the thickness,²² and the maximum ratio of localized to uniform deformation is around 4. This much larger localized deformation w_2 may contribute to the faster gas transport of CO_2 in chiral tubes. In contrast, smaller and uniform deformation in non-chiral tubes may hinder the CO_2 transport. This is directly analogous to driving a screw into wood, where the sharp threads concentrate stress and cause larger localized deformation, displacing material to enable fast forward motion along the axis.

The same mechanism is unlikely to operate for N_2 . No significant molecular spin in N_2 molecules can be introduced by chiral tubes due to the linear molecular geometry. Therefore, thermal energy is always dominant for N_2 molecules, leading to uniform molecule-wall interactions, such that there is no significant spin-enhanced transport in chiral tubes for N_2 molecules, consistent with calculated diffusion rates described above.

Reviewer #2 (Remarks to the Author):

This study investigates a novel diffusion mechanism in chiral hexagonal boron nitride nanotubes (hBNNTs), wherein the chirality of the nanotubes imparts a "rifling" effect that aligns and enhances CO₂ diffusion along the nanotube axis. Through first-principles simulations, the authors demonstrate that CO₂ diffusion in chiral hBNNTs can significantly surpass that in non-chiral nanotubes of similar diameters. The study also evaluates the feasibility of a membrane-based separation system leveraging this directional diffusion to improve CO₂/N₂ selectivity. While the concept is innovative, some inconsistencies in the data and diffusion trends raise questions regarding the fundamental transport mechanisms at play. Specifically, the diffusion enhancement and adsorption behavior require further scrutiny to establish their validity and consistency across different nanotube structures.

We thank the reviewer for his/her assessment of our manuscript and comments/suggestions.

1. The analogy between CO₂ diffusion and the movement of a bullet is highly misleading. In ballistics, the orientation and rifling effects of a bullet influence the flow pattern of air surrounding it, thereby altering the frictional forces that are crucial for stability and accuracy.

Friction is the dominant factor affecting a bullet's trajectory, making rifling essential. However, in gas diffusion at the molecular scale, there is no comparable frictional force acting on CO₂ molecules. The orientation and spin of CO₂ do not produce the same stabilizing or accelerating effects seen in bullets. Instead, the movement of CO₂ is dictated by adsorption-desorption dynamics, interactions with the nanotube walls, and random thermal motion. The bullet analogy is simply wrong.

Response 1:

While macroscopic properties do indeed describe a bullet's stability, these properties themselves result from the atomic-scale properties that are ultimately responsible for the molecule's stability. At atomic scales, macroscopic properties such as friction cannot be applied, regardless if it's bullets or CO₂. Furthermore, drawing an analogy between two systems exhibiting the same behavior does not imply the same underlying mechanisms are responsible. Analogies are routinely made help readers understand the observed behavior and we feel that this analogy, while not perfect, represents an easy visual for readers to follow and we feel that it remains pertinent to the presentation of the molecular-level phenomenon that we believe to have discovered.

2. The study asserts that chiral hBNNTs enable a "non-Knudsen" diffusion mechanism with enhanced directionality. However, Table 1 shows that for the non-chiral (9,0) nanotube, the self-diffusivity of CO₂ is lower than its Knudsen diffusivity ($D_{\text{self}}/D_K = 0.82$), whereas for chiral tubes, the ratio is higher. Given that Knudsen diffusion should generally dominate at these length scales, how do the authors justify the claim that the rifling-induced spin is the primary factor in enhancing CO₂ transport, rather than simple variations in adsorption strength or tube diameter effects?

Response 2:

This is a great point raised, and we ask the reviewer to read our response to your comment 4. The tube diameter doesn't have a clear trend in diffusivity. This underlines a subtle interplay between the radius (which directly affects the hyperloop effect) and the topology (i.e., chirality, which affects the collision) which we hope to have made clearer.

3. The manuscript primarily focuses on CO₂ diffusion, with minimal discussion on N₂ diffusion behavior. The data suggest that N₂ diffusion is largely unaffected by the chiral characteristics of the nanotubes, yet no detailed explanation is provided. What is the underlying reason for this

discrepancy? If chirality significantly influences CO₂ transport, why does it not induce a similar effect on N₂ diffusion? Is this due to differences in molecular interactions with the hBNNT surface, differences in quadrupole moment, or another factor?

Response 3:

Thank you for this suggestion. We have now calculated the diffusivity of N₂ for all tubes to make a more thorough comparison. Discussion of new data and comparison with CO₂ have been implemented in the revised manuscript. Although N₂ and CO₂ show certain difference in the diffusivity trend versus the tube radius (due to their nature), chirality does exhibit similar effects on the transport, that is the enhancement of the diffusivity: in tubes of similar radii, N₂ diffuses faster in chiral tube, for example, (5,5) and (6,4) have similar radii but N₂ diffuses faster in (6,4); likewise, (7,3) and (9,0) have similar radii but N₂ diffuses faster in (7,3).

4. The manuscript does not provide sufficient details on the adsorption behavior of CO₂ and N₂ within the nanotubes. If chirality significantly influences CO₂ diffusion, it is reasonable to expect a similar effect on CO₂ adsorption. Do the adsorption energies or molecular orientations of CO₂ inside chiral nanotubes differ from those in non-chiral nanotubes? Furthermore, the authors should clarify whether N₂ adsorption is similarly affected by chirality. If N₂ diffusion remains relatively unchanged, does this imply that the adsorption characteristics of N₂ are also independent of chirality?

Response 4:

We thank the reviewer for this comment, however we cannot link adsorption strength to diffusion rate. As pointed out in our original submission, the binding of CO₂ into hBN tubes is mainly attributed to Van der Waals interactions, the tube radius (thus, the density of N and B atoms in the surrounding of CO₂) plays a key role. Since the radius of the tube is calculated through the chiral indices (as shown in our manuscript, $r = \frac{a_0}{2\pi} \sqrt{m^2 + nm + n^2}$ with (m,n) being the chiral indices and $a_0 = 2.50 \text{ \AA}$) we then can say the adsorption energy depends on chirality as well. However, we should point out that, the chiral – nonchiral comparison is more relevant if two tubes have very similar radii. For example, in (7,3) vs (9,0) (3.54 vs 3.58 Å) comparison, the chiral tube (7,3) has a slightly higher adsorption energy than the non-chiral tube (9,0).

As such, we do not feel that there needs to be any change to the manuscript regarding this comment.

5. The authors claim that the chiral nanotubes cause CO₂ diffusivity to deviate significantly from Knudsen diffusivity. However, the simulation results presented in Table 1 indicate that even non-chiral nanotubes, such as (5,5) and (9,0), exhibit diffusivities that differ notably from Knudsen predictions. This suggests that factors other than chirality are influencing diffusion behavior. Could the observed deviations be attributed to variations in pore size, surface roughness, or adsorption effects rather than the rifling mechanism?

Response 5: The reviewer posits good explanations for consideration. As we pointed out the “hyperloop” phenomenon in the main text (indicated by the intersection angle between the axis of a gas molecule and the axis of the pore channel) plays a role in molecules’ transport. Based on this, the CO₂ diffusivity in tube (5,5) will be higher than those in tubes (9,0) and (7,3) due to a smaller intersection angle. However, the chiral tube (7,3) has a larger CO₂ diffusivity which is ascribed to the extra enhancement due to the chirality. As noted, CO₂ diffusivity is affected by different factors, and the chirality effect may be overlapped with other effects such as surface roughness. But we observed a correlation between the CO₂ diffusivity and the chirality of the nanotubes which may be explained by the additional freedom to spin (rotation) of CO₂ in chiral nanotubes. To further understand the role of rotation, we calculated the diffusivity of rigid CO₂ (straight geometry, with no rotation about its own OO axis) in (7,3) and (9,0) and found significant lower values (Please see our response to comment 1 by Reviewer 4). The enhancement of CO₂ diffusivity due to the flexible (bent) geometry (which leads to a rotation/precession), which underlines the importance of spin/rotation on the rates of diffusion observed in these calculations. We again highlight that previous studies of CO₂ diffusion have shown that flexible molecules diffuse faster than rigid ones as flexible molecules can adapt their shape to reduce steric hinderance, facilitating transport, which we describe in more detail in response to Reviewer 3’s comment regarding rigid VS flexible molecules. <https://pubs.acs.org/doi/pdf/10.1021/acs.jced.9b00006>).

Please also see our response to comment 3 by Reviewer 1 in which shows our additional study of mechanical effects on CO₂ Knudsen diffusivity deviation from calculated diffusivity.

6. The study presents adsorption energy trends indicating that the (7,3) nanotube has the strongest CO₂ binding energy (-57 kJ/mol). However, strong adsorption is typically associated with slower diffusion due to increased residence time. How do the authors reconcile this contradiction?

Response 6:

We appreciate this comment and apparent conundrum. Sorption is a thermodynamic phenomenon independent of time while diffusion is a transport phenomenon dependent on time. Just by observing diffusion, one cannot make any conclusion about the strength or weakness of sorption. If anything, stronger adsorption leads to a *higher* concentration of gas molecules in the membrane which then leads to a *higher* flux through the membrane.

The adsorption energy shows the binding strength of CO₂ (with the gas-phase as the reference state) and hBN while the diffusion of CO₂ inside the tubes shows how rough the potential energy surface that CO₂ experiences. Scheme R1 below shows that in the case of (7,3) tube, the energy changes more significant when CO₂ moves from the gas phase to the tube pore than in the case of (9,0) tube. After this, however, the “barrier” for CO₂ diffusion in the pore is lower in (7,3) than in (9,0) due to the arrangement of electron clouds (and other factors) as described in our work.

We have added the figure to the revised SI that we hope will provide visual aid that can help clarify adsorption and diffusion in chiral vs a non-chiral tube.

Scheme R1 Potential energy surface upon CO₂ adsorption and diffusion

Reviewer #3 (Remarks to the Author):

The authors investigate the rifling transport of CO₂ using molecular dynamics, aiming to find applications in the field of gas separation. As highlighted by the authors, molecular transport across nanotubes has been a focal point of interest for uncovering novel mass transport phenomena in confined spaces. In this study, the authors delve into understanding the rifling transport of CO₂ within chiral hexagonal boron nitride nanotubes. Although the concept is highly interesting, the analysis remains superficial, and the statistics presented appear

questionable. The authors are strongly encouraged to provide further details on both the systems utilized and the statistical methods applied to ensure the robustness and credibility of their findings. A comparison with current technologies is highly beneficial to engage a broader readership while carefully emphasizing both the limitations of the study and their future research plans.

Response: We thank the reviewer for his/her assessment of our manuscript. For the systems utilized and the statistical methods – related concern, we have done our best to provide longer and more accurate calculations in addition to calculations on rigid VS bent CO₂ molecules. With so many constructive comments, we have provided a complete overhaul of the work and have since strengthened the conclusions made in the initial submission.

Per the comment that there are no “current” technologies to compare to, the reviewer is correct. We wish there was something that we could compare to, but this newly discovered molecular-level phenomenon does not have a direct comparison outside of our discussions of the prior art where we have linked the rate of the diffusion of water in chiral VS non chiral carbon nanotubes. As such, we feel that our conclusions and work represent something that is truly novel and worthy of publication in Nature Communications.

Major Comments

1. The authors report a very interesting CO₂ transport phenomenon across chiral hexagonal boron nitride nanotubes. However, the chosen system and the presented statistics are elusive and unclear to the reader. A commonly employed strategy for simulating membrane-based processes involves using a pressurized feed gas that undergoes separation at a selective interface (in this case, the nanotube film). By recording the number of escaped molecules, it is possible to estimate gas permeance and selectivity. According to the methods section, there are serious concerns regarding the statistical reliability of the study, given the very short simulation times, which are limited to a maximum of 70 ps. Furthermore, the absence of a sufficiently large simulated system to provide robust permeation statistics significantly undermines the validity of the findings. Without strong statistical proof of the system, I cannot recommend its publication in Nature Communications.

Response 1:

We thank the reviewer for these comments and have revamped our calculations to provide the requested statistical rigor and improve the presentation of this work.

In regard to the first comment in relation to the chosen systems: We used adsorption energies of CO₂ in several tubes as the selection criterion, as already detailed in the manuscript as this is a first principles -based study. As the main goal of this work was to discover and understand how chirality affects transport behavior of CO₂ in hBNNTs, and we used periodic models with large unit cells as they can represent the very long transport channel for the molecule.

For the comment related to needing better statistics: To have better statistics, we have carried out molecular dynamics simulations using a new methodology (machine learning interatomic potentials). These new statistics are derived from 10 ns trajectories fully support our conclusions in the original manuscript. To further address this comment, we have implemented these new statistics and detail the methodology of these new simulations in the revised manuscript. Further, we also calculated the diffusivity of N₂ in all tubes and compared to that of CO₂ to ensure that all calculations have this level of rigor.

2. The gas force fields discussion is missing in the method sections. It is particularly important to report both non-bonded and bonded interactions while CO₂ bond flexibility is at the heart of this work.

Response 2:

In this work we did not use classical force fields. In the original manuscript, we used ab initio molecular dynamics (AIMD) simulations, in which the bonded (intramolecular) and non-bonded interactions are described at the *first principles* (density functional theory) level. In the revised manuscript, we use machine learning interatomic potentials (trained on the AIMD data), which we detailed in the comment above and in the revised manuscript.

We have no change in our manuscript regarding this comment.

3. The potential of mean force (PMF) would be ideal to reinforce the claims of the manuscript. By sampling the free energy profile across the different nanotubes, one can assess precisely which nanotube would lead to a faster permeation rate. Authors are highly encouraged to run analysis through selective nanotube channels to reinforce the discussion about permeation.

Response 3:

Thank you for the suggestion. In the revised manuscript we have added the PMFs with the collective variable being the z component of the nitrogen-oxygen distance (two electronic-rich atoms), which is part (k-o) of the updated figure 3 in the revised manuscript (see, Figure R1). The PMFs clearly show that CO₂ experiences rougher free energy surfaces in non-chiral tubes than in chiral tubes.

Figure R2. Updated figure 3 in the revised manuscript

Minor comments:

4. Figure 3 caption is quite confusing for the reader.

Response 4:

The updated caption now reads (according to the updated figure)

“Transport of CO₂ is connected to electronic properties of hBNNTs: (a-e) Visualization of electron density (iso-surface value = 0.3 au); (f-g) xy-average electrostatic potential along the tube direction; (k-o) potential of mean force (PMF) with the collective variable being the z-component of the O-N distance (O: oxygen of CO₂ and N: nitrogen of a hBNNT).”

5. The manuscript would really benefit in clarity by name chiral and non-chiral CNT with distinctive names such as “chi” at the end of chiral CNT names. This would enable smoother discussions.

Response 5:

Thank you for this suggestion. In the revised manuscript, in most cases, we now mention chiral/non-chiral before tube index. We hope that these changes will be easier for readers to

follow when reading this work.

6. The manuscript would benefit from detailed method on how the electrostatic potential was computed.

Response 6:

We thank the reviewer for suggesting inclusion of more information on the calculations. We have added the following to the SI:

“S1.3. Electrostatic potentials

Suppose that z is the tube axis direction, we calculated the xy -plane electrostatic potential for all points within a cylinder of radius R_c centered at the tube axis, Scheme S1, as

$$V(z) = \frac{1}{\pi R_c^2} \int_{x^2+y^2=0}^{x^2+y^2=R_c^2} V(x, y, z) dx dy$$

where $V(x,y,z)$ is the electrostatic potential (calculated using DFT) at point (x,y,z) . $V(z)$ shown in Figure 3, main text, corresponds to $R_c=0.5 \text{ \AA}$.

Scheme S1. Points within the cylinder (dashed lines) of radius R_c centered at the tube axis used to calculate $V(z)$

”

Reviewer #4 (Remarks to the Author):

The authors have likely identified a significant and interesting phenomenon with publishable data. There are however many technical deficiencies in the manuscript that need to be addressed before the manuscript can be more carefully reviewed. The manuscript has some strong aspects including a compelling result and some good art work. It is also scientifically quite sloppy. The manuscript needs a careful makeover and re-review before it can be assessed properly.

1. An issue is that if the authors want to consider the vibrational distortion of the molecule as a critical parameter this cannot be obtained from purely classical dynamics. The nuclear dynamics i.e. vibrations are quantized and distortions would not be expected to be represented in this work using DFT MD. The chirality of the tubes, as is well demonstrated in the animations that are quite nice, defines a preferred direction of diffusion. It is not at all obvious to me that this phenomenon would not occur with rigid a rigid CO₂ model. The idea of the spinning molecule based on the distorted geometry is unconvincing as presented. Indeed, I am not sure at all that is what they are seeing. Checking to see if even an imperfect empirical MD simulation with rigid CO₂ molecules reproduces the phenomena could address several issues including the time scale

concerns in modeling diffusivity. Also, a spinning motion would imply a large moment of inertia with respect to the small vibrational amplitudes and large energies I think.

Response 1:

We thank the reviewer for identifying the potential of this work, but also areas of improvement. There are a lot of individual comments above, so we have broken out detailed responses to each, below.

- a. To the first point *“An issue is that if the authors want to consider the vibrational distortion of the molecule as a critical parameter this cannot be obtained from purely classical dynamics. The nuclear dynamics i.e. vibrations are quantized and distortions would not be expected to be represented in this work using DFT MD.”* It is true that the theoretical framework employed in this study does not capture nuclear quantum effects. However, these effects might be important in H-containing systems like water. Nevertheless, for studying transport of water in nanotubes in recent literature, methods that incorporate quantum nuclear effects like path integral molecular dynamics are rarely used, purely classical dynamics approaches are widely adopted (see for example, <https://www.pnas.org/doi/epdf/10.1073/pnas.2211348119>).
- b. To the second point *“The idea of the spinning molecule based on the distorted geometry is unconvincing as presented. Indeed, I am not sure at all that is what they are seeing. Checking to see if even an imperfect empirical MD simulation with rigid CO₂ molecules reproduces the phenomena could address several issues including the time scale concerns in modeling diffusivity. Also, a spinning motion would imply a large moment of inertia with respect to the small vibrational amplitudes and large energies I think”.* We have carried out MD simulations for a rigid CO₂ molecule in hBNNTs per the suggestion. Two approaches were adopted: Machine learning interatomic potential MD and classical force field MD.

MD based on machine learning interatomic potentials

To address your (and other reviewers') time scale concern for better statistics and longer trajectories, we used AIMD trajectories as datasets to train machine learning interatomic potentials (please see updated method section in the manuscript), which have the AIMD accuracy but offer much faster simulations (please also see our response 1 to reviewer #3). In addition to the case of flexible CO₂, a 70-ps AIMD trajectory for CO₂ in (7,3), in which CO₂ was constrained to a rigid, linear geometry (matching its equilibrium structure at 0 K), was generated. We used it for datasets to train a MLIP. Please see Method section in revised manuscript for more details.

10-ns statistics allowed to compare rigid CO₂ with flexible CO₂ in (7,3). Table R1 below shows that flexible CO₂ diffuses faster than rigid CO₂. This can be interpreted as the following. For flexible CO₂, bending and stretching allow energy to be more effectively redistributed among vibrational, rotational and translational modes. Rigid CO₂ has only translational and (two) rotational degrees of freedom which can limit how energy is transferred during collision with tube walls. Under confinement, flexible CO₂ can adapt its shape and reduce steric hinderance, facilitating faster diffusion. This is opposite to rigid CO₂ which experiences more resistance in the tube.

Table R1. Diffusion of flexible and rigid CO₂ in (7,3) from MLIP MD simulations. The [100,150] ps interval of the MSD was used to evaluate the diffusion coefficient.

	Flexible CO ₂	Rigid CO ₂
Mean square displacement vs time		
Diffusion coefficient (10 ⁻⁹ m ² /s)	5521	2443

MD based on classical force fields

In an alternate, yet complimentary approach we compared the diffusivity of rigid and flexible CO₂ molecules in (7,3) and (9,0), using classical force fields MD. Again, simulations were carried out using LAMMPS. We adopted the Tersoff potentials for hBN bonded interactions (<https://journals.aps.org/prb/abstract/10.1103/PhysRevB.84.085409>) and the TraPPE rigid/flexible model for CO₂ (<https://pubs.acs.org/doi/full/10.1021/je5009526>). The nonbonded interaction between CO₂ and hBNNTs was described by electrostatic and Lennard Jones (LJ) potentials, with the partial atomic charges and LJ parameters for hBN (<https://pubs.rsc.org/en/content/articlelanding/2015/ra/c4ra17048b>) and the LJ parameters for CO₂ (<https://pubs.acs.org/doi/full/10.1021/je5009526>) from the literature. 10-ns trajectories were used to calculate the diffusivity. Table R2 summarizes results. Consistent MLIP MD studies, classical force field MD simulations show that flexible CO₂ diffuses faster than rigid CO₂.

Table R2. Diffusion of flexible and rigid CO₂ from classical force field MD simulations. The [100,150] ps interval of the MSD was used to evaluate the diffusion coefficient.

	Flexible CO ₂	Rigid CO ₂
Mean square displacement vs time		
Diffusion coefficient (10 ⁻⁹ m ² /s)	20165 (7,3) 3043 (9,0)	12819 (7,3) 930 (9,0)

We then did a detailed literature review and found prior art that has reported that, flexible CO₂ diffuses faster than rigid CO₂ in multicomponent systems (<https://pubs.acs.org/doi/pdf/10.1021/acs.jced.9b00006>), which we feel further strengthens the data and our conclusions.

We again thank the reviewer for making this comment as we now have two different levels of theory show that both show that flexible CO₂ (bent geometry) diffuses faster than rigid CO₂. As such, with the prior art supporting similar behavior, we feel that in these tubes, the distorted/flexible CO₂ can adapt its shape and reduce steric hinderance, facilitating transport as detailed in the revised manuscript.

2. It would be helpful if the authors explained the very different binding mechanism for (9,0) and all the other systems as shown in the SI radial distributions. The binding energies are comparable, but this is clearly quite distinct.

Response 2:

(Please also see our response 6 to Reviewer 2) Although the binding energies are comparable (due to the van der Waals nature of interaction and comparable radii), the arrangement of electron “clouds” (at N atoms) in the tube pores changes from one tube to another (also the electrostatic potential). The arrangement of these clouds in (9,0) makes it more difficult for CO₂ (and its orbitals) to advance along the tube direction, as shown in Figure 4.

We have no change in our manuscript regarding this comment.

3. The authors state “CO₂ is very close to the wall in tubes with small radii, resulting in large Pauli repulsions and weakening the CO₂-hBNNT bonding.” There is little evidence of this in Fig. 2. The confined tubes have strong comparable net binding energies and little evidence of weakened bonds. I am not sure what the point is here and it distracts from the basic point that confinement greatly enhances favorable dispersion interactions, especially compared to flat surfaces. It seems these strong interactions might be enhancing directional diffusion in the chiral tubes?

Response 3:

We thank the reviewer for this comment and provide clarification on the two points within.

Figure R2 (Fig. 2 in the manuscript) shows the decreasing trend for the radius: (7,3) > (6,4) > (5,5) > (8,0) and the decreasing binding energy trend (more negative means higher binding energy): (7,3) > (6,4) > (5,5) > (8,0). Keep in mind that the CO₂-hBN interaction is mainly Van der Waals, the equilibrium distance is large (for example, the one for CO₂ – hBN surface is 3.55 Å as provided in the manuscript). A distance less than 3.55 Å would trigger Pauli repulsions. This explains why small tubes ((6,4) – 3.47 Å and smaller) have low adsorption energy.

For the other point the reviewer raised, the correlation between directionality and binding energy is seen in some cases for example, (7,3) and (9,0), but is not seen in some other cases, for example (7,4) and (9,0). So, we do not conclude that strong interactions might be enhancing directional diffusion in the chiral tubes.

No change was made regarding this comment.

Figure R3. Binding energy of CO₂ in hBNNTs vs tube radius

4. The authors follow up by stating “Having identified tubes with optimal radii and adsorption energies, we focused on probing CO₂ diffusion in tubes with near identical radii, but with different orientations of the hexagonal patterns.” In what sense is this optimal? It is far from obvious that the strongest binding energy is best for diffusivity. The connection needs to be made to state this.

Response 4:

The reviewer is correct that specific radii are not optimal, and we apologize for any insinuation that they were. In this first principles study, we just chose 3 chiral tubes and 2 non chiral ones all of which offer *good* binding of CO₂ to make as much of an apples-to-apples comparison to assess the effects of chirality. In the context of membrane-based separation, the adsorption of gas-phase molecules into the membrane is a critical first step that directly influences the overall separation performance. If this initial adsorption is energetically unfavorable – commonly referred to as an uphill process, where the free energy of adsorption is positive, fewer molecules will enter the membrane, thereby reducing the loading of the membrane, ultimately lowering the efficiency of the separation. Therefore, ensuring a thermodynamical favorable (downhill) adsorption process is essential for achieving effective separation performance.

We have changed the sentence “Having identified tubes with optimal radii and adsorption energies, we focused on probing CO₂ diffusion in tubes with near identical radii, but with different orientations of the hexagonal patterns” to “*Having identified tubes with large binding energies, we focused on probing CO₂ diffusion in such tubes with similar radii, but with different orientations of the hexagonal patterns.*” We hope this change reflects the clarification that the reviewer was asking for.

5. This is from the text:

Static density functional calculations for binding energies

The binding energy of a (confined) CO₂ molecule on a BNNT is calculated as

$$\Delta E = E(\text{CO}_2 - \text{BNNT}) - E(\text{CO}_2) - E(\text{BNNT}),$$

in which the energy E of each system was calculated at the density functional level provided in S1.2.

This is from the SI:

S1.2. Static density functional calculations for binding energies

The binding energy of a (confined) X (X=CO₂, N₂) molecule a hBNNT was calculated as

$$\Delta E(X) = E(\text{CO}_2 - \text{hBNNT}) - E(X) - E(\text{hBNNT}),$$

in which the energy E of each system was calculated at the density functional level provided in Materials and Methods (main text).

So how was it calculated? This is sloppy.

Response 5:

There were redundant/inconsistent details in original manuscript, thank you for pointing this out. The binding energy method section in the revised manuscript reads as

“Static density functional calculations for binding energies

The binding energy of a (confined) CO₂ or N₂ molecule (mol) in a hBNNT was calculated as

$$\Delta E = E(\text{mol} - \text{hBNNT}) - E(\text{mol}) - E(\text{hBNNT}),$$

in which the energy E of each system was calculated at the density functional level provided above.”

6. A minor point but these are not really AIMD but rather use DFT methods. DFT MD or periodic electronic structure MD are preferred IMO.

Response 6:

AIMD and DFT MD are used interchangeably in the literature. We wish to keep “AIMD” in this work as consistent with many publications in Nature Communications (see, for example, <https://www.nature.com/articles/s41467-025-56222-0>)

7. The text uses free energy estimates that are given but not in any way justified in the SI. This needs to be addressed.

Response 7:

This is an approach for chemical potential of gas phase energy. In the revised manuscript we cited the foundational work by Reuter and Scheffler (<https://journals.aps.org/prb/abstract/10.1103/PhysRevB.65.035406>)

8. I am concerned about the MSD calculations. The authors need to show they are linear. They seem to compare to a more approximate method in the SI (Fig. S2,S3) but only show dynamics over the same short time range. It is not at all clear they are in the linear regime. I believe they are seeing a real phenomenon, but this is a high level journal and the work is not thorough or ostensibly carefully done.

Response 8:

We appreciate the reviewer asking for more details and ensuring the high-quality requisite for journals like Nature Communication. As detailed in previous responses, and detailed in the revised manuscript, we have used molecular dynamics based on machine learning potentials, and statistics was obtained from 10 ns which we feel are long enough to have sufficiently good statistics. As mentioned before, these longer simulations did not change the main findings of our work, rather they have strengthened the conclusions. We have implemented new data in the revised manuscript. In addition, per suggestion by Reviewer #1, we further calculated the diffusivity of N₂ in all tubes. CO₂ - N₂ comparison has been elaborated in the revised manuscript.

9. The distribution of angles is given in the SI but this is not tied to any proof of a rifling effect in the text. As I said I am not convinced of this mechanism but if they want to claim this as the reason it needs to be justified.

Response 9:

We showed the angle between the tube and molecular (OO) axes, which was tied to the hyperloop. The closer to 0°, the stronger the alignment in hyperloop would happen. In the original manuscript, we used this to show that the enhancement would have been faster for smaller angle systems, but it is not always the case, suggesting other factors at play.

10. The videos show a regular librational motion that in the diffusive (7,3) is apparently coupled to directional transport and the higher amplitude (9,0) seems to be normal diffusion. This coupling of motions does seem to be at least a part of the observed directional diffusion mechanism. Is this what the authors mean by hopping? The S6 figure does not show anything exceptional that correlates.

Response 10:

By hopping between two B sites, which is characterized by the change in cylindrical coordinates. The hopping patterns observed in the five hBNNTs align well with the diffusivity trend.

I could continue but the authors need to take some time to better present this very compelling and interesting work in a more carefully.

We thank the review for the praise and challenge. We hope that these revisions and extra work including a section “Mechanics aspects of nano-rifling and spin-enhanced transport” sufficiently address these highly constructive comments and strengthen the conclusions and presentation of our findings. We look forward to a second round of comments.

Dear Reviewers

Thank you again for the opportunity to have a second round of revisions. We appreciate you providing objective feedback and suggestions for us to refine and improve the presentation and the novelty of these new findings. We have focused on revamping the presentation performed new simulations and analysis that have helped shed light on the data and slightly shifted our conclusions. Below, we respond to comments to the best of our availability. We uploaded a clean version of the manuscript in addition to a red-marked version to see the location of the revisions. We have also provided 4 supplementary videos that shows the precessional motion of the CO₂ in the 9,0 (non-chiral) and the 7,3 (chiral) tubes as a visual aid. We hope that these revisions strengthen the work and presentation and highlight a unique and promising new type of diffusion that may help the field unlock new and more efficient separations.

Thank you again for your time and consideration,

Dave, Manh, Abhoyjit, Jay and Jian

REVIEWER COMMENTS

Reviewer #1 (Remarks to the Author):

Through simulations, this work unveils an interesting transport mechanism for CO₂, i.e., direction-specific diffusion. With appropriate size of hexagonal boron nitride nanotubes, CO₂ permeability of 1.35×10^7 Barrer and CO₂/N₂ selectivity of 170 were achieved. This concept is novel. My other comments are shown below.

1. Knudsen diffusion was calculated inside the tubes. However, the tubes have sizes (0.6-0.8 nm) close to the CO₂ molecule (0.33 nm). Should other diffusion mechanisms be considered, such as blocking or surface diffusion? The blocking model has also been used for inorganic molecular sieve membranes.

Knudsen diffusion provides an ideal limiting case when elastic collisions between molecule and pore wall dominate the overall transport mechanism. A comparison to Knudsen is a reasonable way to determine whether such an ideal case exists. If not, then other transport mechanisms should be considered. The blocking model may not be a good one to describe the process because no pore blocking nor size exclusion is happening in the case of CO₂/N₂ penetrating the hypothetical membrane.

2. The permeability was estimated based on a porosity of 0.9069, which seems to be very high. Is it possible to estimate a more real porosity based on other nanotube studies such as CNTs?

Thank you for the suggestion. We calculated the permeability of a practical membrane (porosity 0.15) as the lower limit case and compared with that of the (7,3) hBNNT. The results shown in Figure 5 indicate that the lower limit case of permeability is about 1/15 that of the hypothetical (7,3) hBNNT membrane. Carbon nanotube porous network also has high porosity (>0.9 , Adv. Energy Mater. 2019, 9, 1900914). For some applications of using carbon nanotubes in a polymer matrix, a 0.2 porosity was reported (Membranes 2020, 10 (10), 273). This porosity was also covered between our upper and lower limits shown in Figure 5.

3. For the membrane calculation, what is the membrane thickness?

We reported the calculated permeability which is a material property independent of membrane thickness. We did not report permeance, which can be calculated by dividing the permeability by the membrane thickness. A possible thickness of the hypothetical membrane is the thickness of a single layer of hexagonal BN.

Reviewer #2 (Remarks to the Author):

The central concept of “molecular spin” and “rifling motion” of CO₂ in chiral hBNNTs is physically questionable and insufficiently supported. According to group theory, a linear centrosymmetric molecule such as CO₂ does not possess a genuine rotational degree of freedom about its molecular axis, and thus cannot sustain the type of axial spin the authors propose.

The “spin” observed in their simulations appears to arise only from slight bending distortions of the molecule under confinement, rather than from any well-defined angular momentum transfer. Furthermore, the analogy to rifled bullets is physically misleading, since diffusion in nanopores is dominated by Brownian dynamics and interaction potentials rather than torque-induced rotation. While confinement and chirality may indeed bias molecular orientation and diffusivity, the claim of a “nano-rifling” mechanism is very misleading, as it currently lacks experimental validation and risks overstating the physical plausibility.

I therefore cannot support publication of this work unless such terminology is removed.

We thank the reviewer for the comments and have since updated the entire manuscript accordingly. We have removed all references to the analogy of rifling, ballistics, and spinning and have focused the discussion to the desire to design a material that could achieve near-perfect diffusion without collisions to the tube wall resulting in directional changes and slower diffusion.

We also note that the reviewer was correct in that the molecule is not “spinning” as our earlier analogy. In this revision, we have updated our analysis and discussion to detail how the CO₂ molecules confined inside the tube are slightly distorted and end up undergoing a type of molecular precession, which is a rare, but known phenomenon. The motion occurs because the CO₂ distorted inside the nanotubes gains a weak dipole which moves in response to a continuous rotating electric field, *i.e.* the nitrogen orbitals along the chiral index. We have new analysis demonstrating a precession of CO₂ and have updated the discussion accordingly,

Outside of a substantial rewrite of the introduction and conclusion per the reviewer’s request, we highlight the following analysis, and discussion has been added to the paper, along with new supplemental figures and data that we feel addresses the reviewer’s main concerns:

“Precession-like behavior of CO₂

Figure 4 (a) shows the precession of a spinning top, with its two different co-occurring rotations: (1) spin, or the fast rotation about its own axis, (S) and (2) the slow rotation of the spin axis about an external axis (P). Linear CO₂ is less diffusive than flexible CO₂, suggesting that molecular rotation may play an important role in this process. In this section, we analyze the rotation of CO₂ (specifically the C atom) about its own OO axis, facilitated by its bent geometry (Figure 4 (b)), and

the rotation of the OO axis about the tube axis direction (\vec{z}), facilitated by the tilting of OO with respect to (\vec{z}) (Figure S9). Here we focus on the two extreme cases: (9,0) and (7,3) tubes.

Figure 1 (a) Precession of a spinning top; (b) Two different rotations of CO₂; (c) Frequency of rotation of CO₂ in hBNNTs.

Movie S1-2 highlights both the slow, conics-like motion of the OO axis about the tube direction and the fast rotation of the molecular plane. The calculated S and P rotation frequency spectra (see Section S 1.5 for additional details) are shown in Figure 4 (c). In the two tubes, what we define to be a precession frequency is much lower (about two orders of magnitude) than the spin frequency, making it similar to the extremely slow precession trend of well-known objects such as Earth. To further understand the precession-like behavior of CO₂, we calculated the torque of a rigid, bent molecule in the two tubes (Figure S10). While this artificial geometry is not physically accurate, it can be useful for interpreting the precession and the role of molecule-tube interactions in the absence of internal degrees of freedom. The z component of the torque (τ_z), which makes the molecule rotate about the z direction, is higher in the (7,3) than in (9,0)—consistent with the trend of f_P in these tubes. While the precessional rotation in (9,0) may not help CO₂ reduce collisions with the tube wall, Figure 3(k), the asymmetry of the potential may help in the (7,3) tube.”

As such, we thank the reviewer for being firm in their suggested update of the draft as the conclusions of the paper remain the same, however the type of motion we feel is now more appropriately defined as it is consistent with known physical phenomena. We hope the other revisions throughout the draft have addressed the reviewer’s comments and recommendations.

Reviewer #3 (Remarks to the Author):

The authors have addressed my comments.

We thank the reviewer for their assessment and time for their review.

Reviewer #4 (Remarks to the Author):

I am happy overall with the response to all the reviewers and the thorough modifications. I will ask that they modify some language here (see below) and their response is actually wrong. As they have established very nicely, the chiral transport depends on the molecular distortion of the CO₂ molecule (the CO₂ bend freq, us 667 cm⁻¹ e.g.). These bends and such are quantum mechanical vibrations at multiples of thermal energy. Thus, their interaction, distortions, vibrational amplitudes etc. are not at all described in DFTMD or using empirical potentials. This is simple physics. What is established is that the distortion couples to the chiral PES and changes the dynamics. I see no reason this won't carry over to the true dynamics that is not easy to simulate but mostly for technical reasons. So I strongly suggest the authors briefly talk about this issue and maybe even give the ratio of the amplitudes for say the quantum harmonic bend to the classical result for the relevant temperature and that suffices with some logic as I outline here. It is an interesting result and I complement the authors for taking the reviewers seriously.

We thank you for your assessment of our previous responses. In the following, we briefly address the comment.

We assume the bending coordinate q as a harmonic oscillator

$$V(q) = \frac{1}{2}kq^2$$

where the spring constant $k = \mu\omega^2$ is expressed in terms of frequency ω and effective mass μ .

Classically, the energy is given by

$$E_{\text{classical}} = \frac{1}{2}kq_{\text{classical}}^2 = k_B T$$

thus,

$$q_{\text{classical}}^2 = \frac{k_B T}{k} = \frac{k_B T}{\mu\omega^2}$$

For a quantum harmonic oscillator, the expectation value of q^2 ,

$$q_{\text{quantum}}^2 = \frac{\hbar}{2\mu\omega} \coth\left(\frac{\hbar\omega}{2k_B T}\right)$$

Finally,

$$\frac{q_{\text{classical}}}{q_{\text{quantum}}} = \sqrt{\frac{2k_B T}{\hbar\omega} \frac{1}{\coth\left(\frac{\hbar\omega}{2k_B T}\right)}}$$

Given $\omega = 667 \text{ cm}^{-1}$, this ratio of amplitude is 0.76 if $T=300 \text{ K}$.

Thus, the classical approach, although underestimating the amplitude, fairly describes the bending of CO_2 . This suggests that the bending (thus, rotation effects) is more likely to be driving this phenomenon.

a. To the first point “An issue is that if the authors want to consider the vibrational distortion of the molecule as a critical parameter this cannot be obtained from purely classical dynamics. The nuclear dynamics i.e. vibrations are quantized and distortions would not be expected to be represented in this work using DFT MD.” It is true that the theoretical framework employed in this study does not capture nuclear quantum effects. However, these effects might be important in H-containing systems like water. Nevertheless, for studying transport of water in nanotubes in recent literature, methods that incorporate quantum nuclear effects like path integral molecular dynamics are rarely used, purely classical dynamics approaches are widely adopted (see for example, <https://www.pnas.org/doi/epdf/10.1073/pnas.2211348119>). b. To the second point “The idea of the spinning molecule based on t...”

We thank the reviewer for the valuable suggestion and agree that these effects are valuable to consider in future studies. Per comment b being truncated, we have tried our best to address it, but we feel there are no further changes required in the revised manuscript.

In all, we thank the reviewers again for the constructive feedback and we have done our best to revise the paper. We hope that these revisions and new additions help strengthen the conclusions, the presentation, and the impact of the paper.

RESPONSE TO REVIEWERS' COMMENTS

Reviewer 4:

The authors have made progress addressing my main concern. Still, I think some appropriate language and or calculations need to address the issue in the manuscript.

The basic idea is that a purely ground state dominated quantum mechanical bending mode of a linear molecular couples to an external potential (provided by the material) in such a way that it bends driving a unidirectional diffusion during its dynamics. They have strong classical evidence as they turn up the frequency (rigid) and eliminate the behavior by reducing the coupling.

I think I understand the confusion. One might think because we could say optimize the molecular structure in the material and obtain forces and repeat and get a series of confirmations that all seem reasonably obtained. In the paper they use a finite temperature empirical potential, but similar considerations arise. I think this is why they keep wanting to ignore quantum mechanics and feel justified.

Consider though as the frequency of the mode increases the amplitude in QMs is not temperature dependent and only relatively weakly indirectly frequency dependent through the dependence of the ground state wavefunction of the potential. Further, this all depends on the coupling to the external potential. The linear molecular in the material will not have pure rotations or vibrations, the modes mix giving the vibrations rotational character that is what presumably causes the effect they see from this perspective. Still the molecule must bend to get the coupling required I believe and this mode is likely to be high frequency and quite quantum mechanical. Some of this might be sorted out by getting the normal modes of the system for a few conformations. As I said last round the classical effect is likely to be preserved even in the real world. But it is up to the authors to explain this in the manuscript in my opinion. It is a requirement to be in a good journal that the physics is well explored. It is a tricky situation and perhaps I am missing something.

As for the previous response:

"= 667 cm^{-1} , this ratio of amplitude is 0.76 if $T=300\text{ K}$. Thus, the classical approach, although underestimating the amplitude, fairly describes the bending of CO_2 . This suggests that the bending (thus, rotation effects) is more likely to be driving this phenomenon. ." This is not how dynamics works coupling to a QM degree of freedom. With some effort the suggestion and analysis has value but it is not "fairly described" as a result of the amplitudes being similar. This is bad. You can't just ignore QM degrees of freedom that are centrally involved in the phenomena of interest because the codes you use don't handle them or neglects them. It may just be as I suggest that the geometry distortion induced by the material brings in to play the low frequency modes that move the system by a combination of rotation / translation of the molecule in the material system that can be treated classically and then the issue is mostly addressed.

Response:

We thank you for your further assessment and comment. We agree that similarity between classical and quantum vibrational amplitudes does not fully justify a classical description of a high-frequency bending mode. We have revised the manuscript to explicitly distinguish between the intrinsic, ground-state-dominated quantum bending vibration of CO_2 and the dominant environment-driven motions captured in our simulations.

In the revised text, we emphasize that within hBN nanotubes, symmetry breaking leads to mixing of rotational, translational, and vibrational degrees of freedom, producing low-frequency, thermally populated modes with partial bending character that are appropriately described classically. We interpret the observed behavior as arising from these environment-induced motions rather than from direct classical excitation of the intrinsic bending vibration.

The following paragraph has been added to the revised manuscript.

"It is necessary to point out that the intrinsic bending mode of CO_2 lies at approximately 667 cm^{-1} and remains largely quantum mechanical at room temperature, with its amplitude dominated by the ground-state wavefunction. Accordingly, numerical similarity between classical and quantum vibrational amplitudes is not interpreted as evidence that the bending dynamics itself is accurately described classically. In a confined pore, however, symmetry breaking leads to mixing between translational, rotational, and vibrational degrees of freedom, giving rise to low-frequency, thermally populated motions with partial bending

character. These nanoconfinement-induced modes are sensitive to the external potential and are well described classically, and they can indirectly modulate the effective bending geometry and orientation of CO₂ without requiring direct classical treatment of the high-frequency quantum bending vibration.”

As you are aware, the primary goal of the present work is to bring to the separation community’s attention a previously unreported phenomenon, which was discovered by using a reliable and well-established computational protocol, that has potential implications for improving separation efficiency and selectivity. The manuscript already contains multiple lines of evidence supporting the existence and robustness of this effect.

We agree that a deeper exploration of the underlying physics is both important and worthwhile, nonetheless, we believe that such an in-depth analysis merits a dedicated, follow-up study. We would like to emphasize that the present manuscript is not intended to represent the final stage of our investigation, but rather the beginning of a broader research effort. We are currently working with chemical physicists (Mundy and Schenter) in our organization to further elucidate the physical mechanisms underlying chirality-enhanced transport of CO₂ in the hBN nanotubes. Below we provide some of our preliminary analyses that support this ongoing effort. Thank you for your thoughtful comments and constructive suggestions.

Our atomistic modeling results reveal a pronounced dependence of CO₂ transport on nanotube chirality, even for nanotubes with nearly identical radii. Specifically, CO₂ diffuses significantly faster in chiral nanotubes (e.g., 73) than in non-chiral nanotubes (e.g., 90). This difference arises from fundamentally different molecular dynamical behaviors in the two environments. Figure 1 illustrates the probability distribution of the axial velocity of CO₂ molecules inside the two nanotubes. In 73, the velocity distribution is unimodal and strongly skewed toward a single direction, indicating persistent, directional motion along the tube axis. In contrast, 90 exhibits a pronounced bimodal velocity distribution, with comparable populations of positive and negative velocities. This bimodality signifies frequent reversals in the direction of motion, implying that CO₂ molecules repeatedly switch between left- and right-moving states in 90. As a result, net axial transport is substantially hindered compared to the chiral case, where direction changes are rare. Further insight is provided by the spectral density of the velocity autocorrelation function shown in Figure 2. For 90, a distinct peak emerges at a characteristic frequency, which can be associated with transitions between positive and negative velocity states. This feature reflects the underlying oscillatory or back-and-forth motion of CO₂ molecules induced by the symmetric potential landscape of the non-chiral tube. In contrast, the absence (or strong suppression) of such a peak in 73 indicates more persistent correlations in molecular velocity and reduced backscattering, consistent with enhanced diffusion. Together, these results demonstrate that nanotube chirality plays a decisive role in governing molecular transport at the nanoscale. Even when tube radii are nearly identical, chiral symmetry breaking leads to qualitatively different velocity dynamics, suppressing directional reversals and promoting faster diffusion.

Figure 1. CO₂ center of mass velocity distribution

Figure 2. Spectral density of velocity autocorrelation functions

Our data highlight nanotube chirality as a powerful and largely underexplored design parameter for regulating gas transport in nanotube-based membranes and nanofluidic devices, extending well beyond the conventional paradigm of size- or radius-based confinement. By introducing asymmetry at the atomic scale, chirality fundamentally reshapes the dynamical landscape experienced by confined molecules, enabling transport behaviors that cannot be accessed through geometric tuning alone.

More broadly, these results point toward richer conceptual frameworks for understanding and exploiting nanoscale transport. One such framework is nonequilibrium control, in which transport properties emerge from the interplay between structural

asymmetry, molecular fluctuations, and driven dynamics. In this context, chiral nanotubes may act as passive rectifiers of molecular motion, evoking analogies to Maxwell's demon-like mechanisms where information, feedback, and entropy production play a central role. Although no thermodynamic laws are violated, chirality effectively biases molecular trajectories, highlighting how controlled asymmetry can be used to steer transport under nonequilibrium conditions.

Another promising perspective is resonant enhancement, where transport is amplified through frequency-matched interactions between molecular motion and the underlying energy landscape of the confining medium. Chirality may selectively couple molecular velocities to specific vibrational or rotational modes of the nanotube, leading to enhanced persistence, suppression of backscattering, and the emergence of qualitatively new nonequilibrium steady states. Such resonance-like effects suggest that transport can be optimized not only by static structural design, but also by tuning dynamical responses to internal or external driving.

Taken together, these considerations suggest that chirality offers a route toward programmable molecular transport, opening new opportunities for the rational design of membranes, selective separators, and nanofluidic devices that operate far from equilibrium.